# Pricing decision and channel selection of fresh agricultural products dual-channel supply chain based on blockchain

Di Wang[1,2]*, Xiaoyue Tian[1], Mengchao Guo[1]

1 Energy Economics Research Center, School of Business Administration, Henan Polytechnic University, Jiaozuo, China, 2 Taihang Development Research Institute, Henan Polytechnic University, Jiaozuo, China

* wangdi@hpu.edu.cn

**Data Availability Statement:** All relevant data are within the manuscript and its Supporting Information files.

**Funding:** This research was supported by the Henan Provincial Department of Philosophy and

## Abstract

The application of blockchain can effectively improve the efficiency of fresh agricultural product circulation and consumer trust, but it can also increase investment costs. In this context, this paper introduces parameters such as blockchain unit variable cost, the level of blockchain technology investment, and consumer channel preference in two dual-channel supply chain systems dominated by fresh agricultural product manufacturers: online direct sales and distribution. It compares and analyzes pricing and channel selection strategies in both cases of not using and using blockchain. The research shows that when blockchain is used, manufacturer profits are higher in the direct sales model than in the distribution model. Traditional retailers' profits are lower in the direct sales model than in the distribution model. Total supply chain profits are higher in the direct sales model than in the distribution model, and they exhibit an inverted "U" shape as the level of blockchain investment increases. In the online direct sales model, if the blockchain technology unit variable cost is within a certain threshold range, manufacturer profits, traditional retailer profits, and total supply chain profits are all higher than when blockchain technology is not used. In the online distribution model, when the blockchain variable cost and blockchain usage level meet certain conditions, manufacturers, traditional retailers, and online distributors all have higher profits when using blockchain technology than when not using it. This study provides theoretical guidance for the practical application of blockchain technology in dual-channel fresh agricultural product supply chains.

## 1 Introduction

With the widespread adoption of Internet technology and the rapid growth of the online retail market, the convergence of online and offline channels has become increasingly sophisticated [1]. Simultaneously, consumers are showing a growing interest in the freshness, safety, and trustworthiness of product information related to fresh produce [2]. This trend has prompted manufacturers, especially those in agricultural industrialization and professional cooperatives, to expand their presence beyond traditional retail channels by venturing into the online

Social Sciences Planning Project (2023CJJ145) and the Fundamental Research Funds for the Universities of Henan Province (SKJYB2023-13) through grants awarded to DW.

**Competing interests:** The authors have declared that no competing interests exist.

sphere, aiming to enhance the quality of agricultural products and capture a larger market share. High-quality agricultural manufactures such as Anchor, Dole, and Jiawo adopt the online direct sales model, while some small-scale agricultural product manufactures choose the online distribution model. On a global scale, the fresh food e-commerce market is experiencing exponential growth. Several leading fresh food e-commerce companies in the United States have achieved valuations exceeding 10 billion US dollars. China, with a more pronounced urbanization-driven population concentration, offers an even larger potential market space for fresh food e-commerce. According to data from the "Electric Data" database for e-commerce, the transaction volume of fresh food e-commerce in China in 2022 reached 560.14 billion yuan, marking a 20.25% increase compared to the previous year [3]. In general, while China's fresh produce e-commerce may constitute a relatively small share, its growth is rapid and holds significant potential. The sales model based on a dual-channel structure will reduce the sales costs for enterprises and increase their market share, thereby benefiting the business [4]. However, the diversification of dual-channel structures, intense channel competition, and the presence of consumer channel preferences easily trigger "free-riding" behavior and vicious price competition. In addition, consumer channel preferences can also have a significant impact on the management decisions of supply chain enterprises, such as influencing the price competition relationships among businesses [5]. In this sense, it is crucial to study the pricing and channel selection issues of the dual-channel supply chain of fresh agricultural products composed of manufacturers and retailers from a systemic perspective.

The perishable nature of fresh agricultural products and the complexity of standardizing production processes result in freshness degradation and substantial physical losses during the distribution phase [6]. According to the Food and Agricultural Organization of the United Nations (FAO), supply chains contribute to more than 30% of waste, with industrialized Asian countries being a notable exception. In underdeveloped regions such as South Asia and Latin America, food waste generated within the supply chain can reach as high as 50%. These uncertainties not only impact the quality and yield of fresh agricultural products but also affect the overall purchasing experience of consumers, presenting a challenging environment for manufacturers seeking profitability [7]. Furthermore, as products circulate, the information regarding product quality continuously degrades, leading to severe information asymmetry in the product quality supervision process [8]. Consumers in an information disadvantage position are unable to trace the quality of the product, and their rights cannot be adequately protected. In the traditional model, information about product quality traceability comes from manufacturers or third-party enterprises. They may alter product information arbitrarily in pursuit of maximizing profits, making it difficult for consumers to effectively discern product quality. This situation can trigger a "trust crisis" among consumers regarding the information disclosed by companies. In the context of adopting blockchain technology, Leveraging its decentralized, tamper-resistant, and highly transparent characteristics, blockchain technology offers an efficient solution to address the issue of uncertain product quality resulting from information asymmetry within the supply chain. It can also reduce the time required for the movement of goods between upstream and downstream businesses, improve collaborative efficiency, ultimately enhancing the overall customer experience [9]. In recent years, it has garnered significant attention and has found applications in various economic and social sectors, including agriculture, healthcare, and management [10–14]. Prominent companies such as JingDong and Wal-Mart are already harnessing blockchain technology to monitor and record every stage of fresh agricultural product production, thereby ensuring product quality and safety while instilling greater confidence in consumers [15]. Additionally, within the financial sector, blockchain technology facilitates the creation of digital assets, including fungible and non-fungible tokens (NFTs). It also supports various applications such as patent issuance and

biotechnology grants [16]. Therefore, the adoption of blockchain technology represents a pragmatic and essential step for enhancing product quality, meeting consumer demands, mitigating competition between fresh manufacturers and retailers, and achieving the harmonious development of dual channels within the fresh agricultural supply chain [17]. In this context, studying the application of blockchain in the supply chain of fresh agricultural products holds significant theoretical and practical importance. It contributes to enhancing the management decisions in dual-channel supply chains, enriching the practical applications of blockchain, and providing valuable theoretical and practical guidance.

Based on this, in a dual-channel supply chain of fresh agricultural products consisting of one supplier and one retailer, this paper introduces parameters such as the circulation efficiency of fresh agricultural products, variable costs of blockchain units, the level of investment in blockchain technology, and consumer channel preferences. Utilizing the Stackelberg game model, the paper conducts a comparative analysis of pricing and channel selection strategies in the dual-channel supply chain of fresh agricultural products under two scenarios: not adopting blockchain technology and adopting blockchain technology. Specifically, the paper establishes a two-stage Stackelberg game model with manufacturers leading and retailers following, considering the scenarios where manufacturers open online direct sales and online distribution before and after adopting blockchain technology. The four modes analyzed are manufacturers leading in online direct sales, manufacturers leading in online distribution, manufacturers leading in both, and manufacturers leading in neither. Subsequently, the paper analyzes the optimal decisions of supply chain participants under these four modes. By comparing optimal pricing and profits under different modes, the paper dissects the conditions for blockchain technology investment and the selection of dual-channel structures in the dual-channel supply chain of fresh agricultural products. Finally, through numerical analysis, the paper explores the impact of blockchain technology on decision-making in the dual-channel supply chain of fresh agricultural products from four dimensions: blockchain usage level, blockchain variable costs, and consumer channel preferences.

The remainder of the article is structured as follows. Section 2 provides a comprehensive review of the pertinent literature. In Section 3, we establish four distinct dual-channel fresh produce supply chain models. Sections 4 and 5 delve into the exploration of optimal pricing strategies and channel selection decisions within these four models. Section 6 offers an in-depth analysis of how blockchain technology influences optimal decision-making in dual-channel fresh produce supply chains. Moving on to Section 7, we carry out numerical examples and sensitivity analyses to derive valuable managerial insights. Finally, Section 8 concludes the paper and outlines potential avenues for future research.

## 2 Literature review

This paper encompasses three main streams of literature: dual-channel fresh food supply chains, the implementation of traceability systems within fresh food supply chains, and the influence of blockchain technology on supply chain decision-making. We have conducted a thorough literature review and situated our research within these domains as outlined below. Additionally, we have conducted a comparative analysis of pertinent literature, and the findings are presented in Table 1.

### 2.1 pricing decisions and channel selection in dual-channel supply chains for fresh agricultural products

In the research on pricing decisions and channel selection in dual-channel supply chains for fresh agricultural products, Liu Molin, Dan Bin, and others [18] comprehensively consider the

**Table 1. A brief review of the related studies on supply chain management.**

| No. | Sub-Category | Year | Channel structure | Decisions | Methodology | Blockchain-based |
|---|---|---|---|---|---|---|
| 1 | / | 2022 | DC[a] | / | Literature review | NO |
| 2 | / | 2020 | / | / | Conditional Logit model, Mixed Logit model, Latent Class model | NO |
| 6 | Cold Supply Chain | 2023 | / | / | Literature review | NO |
| 7 | Cold and Green Supply Chain | 2020 | DC | 8 | Coordination model | NO |
| 8 | Agricultural Product Supply Chain | 2021 | SC[b] | / | LSGDM | NO |
| 9 | Agricultural Supply Chain | 2021 | / | / | Maturity model | YES |
| 10 | Fresh Product Supply Chain | 2021 | SC | 1, 8 | Stackelberg game | YES |
| 14 | Supply Chain Management | 2023 | / | / | PLS-ANN | TES |
| 15 | Precast Supply Chain | 2020 | / | / | / | YES |
| 18 | Fresh Product Supply Chain | 2020 | SC | 1, 4 | Stackelberg game | NO |
| 19 | Fresh Agricultural Products Supply Chain | 2023 | DC | 1, 4 | Stackelberg game | NO |
| 22 | Fresh Food Supply Chain | 2021 | SC | 6 | Stackelberg game | NO |
| 24 | Fresh Agricultural Products Supply Chain | 2023 | SC | 5, 8 | Stackelberg game | NO |
| 25 | Fresh Agricultural Products Supply Chain | 2020 | DC | 5, 8 | Stackelberg game | NO |
| 27 | Fresh Products Supply Chain | 2023 | SC | 6 | Stackelberg game | NO |
| 28 | Fresh Products Supply Chain | 2022 | SC | 1, 6 | Differential game | NO |
| 29 | Fresh Produce E-Commerce Supply Chain | 2021 | SC | 6 | Stackelberg game | NO |
| 33 | Herbal Medicine Supply Chain | 2021 | / | / | Logistics Model | NO |
| 34 | Fresh Agricultural Products Supply Chain | 2019 | DC | 1, 7 | Stackelberg game | NO |
| 44 | Green Supply Chain | 2023 | SC | 8 | Stackelberg game | NO |
| 46 | General supply chain | 2020 | SC | 8 | Stackelberg game | NO |
| 48 | Agri-food traceability | 2020 | / | / | Literature review | YES |
| 59 | Green Supply Chain | 2020 | DC | 1 | Stackelberg game | NO |
| 60 | Green Supply Chain | 2022 | SC | 1 | Stackelberg game | NO |
| Our study | Fresh Agricultural Products Supply Chain | / | DC | 1, 5, 7 | Stackelberg game | YES |

[a]Channel structure: Single channel (SC), Dual channels (DC).

[b]Decisions: Pricing policy (1), Production policy (2), Inventory Policy (3), Ordering quantity (4), Service level (5), Sales Model Policy (6), Investment Policy (7), Coordination Policy (8).

impact of preservation efforts and service levels on market demand. Through comparative analysis of optimal decisions in centralized and decentralized settings, they derive conditions for implementing "high quality, low price" and "high quality, high price" strategies in the supply chain. Ye Jun and others [19] explore the pricing decisions in the fresh agricultural product supply chain under two scenarios, considering cold chain services as both internal and external parameters. Li Lin and others [20], focusing on fresh retailers operating both online and offline channels, consider consumer channel preferences and study the pricing decisions after adopting the BOPS (Buy Online, Pick Up Offline) service model. Liu Molin and others [21] investigate the optimal pricing decisions when fresh suppliers are responsible for preservation efforts and design a "revenue sharing—two-way sharing" contract.

In addition, with the development of e-commerce, different dual-channel structural models will also have a significant impact on the operation of the supply chain [22–25], The choice of sales models for online channels has become an important decision for manufacturers and a focal point for many scholars. In the context of the supply chain for fresh agricultural products, Tian et al. [26] explored the optimal cooperative model selection for O2O fresh agricultural e-commerce platforms under wholesale and commission models. Zhang et al. [27] considering live streaming for product promotion, investigated the strategic choices of members in the supply chain under three models: online direct sales, online distribution, and online consignment, for fresh agricultural products. Hu et al. [28] studied the sales model selection and pricing strategy of the fresh supply chain composed of suppliers and retailers under the pre-sale model. Zheng et al. [29] discussed the online optimal channel selection strategy considering the preservation efforts of supply chain members. Yang and Tang [30] further explored the optimal channel selection decisions among traditional retail models, dual-channel models, and O2O models.

Existing literature primarily considers factors such as channel selection and consumer preferences when examining pricing and channel decisions in dual-channel agricultural product supply chains. In contrast, this paper integrates the issues of dual-channel supply chains for fresh agricultural products with blockchain technology. It takes into account the impact of factors such as the circulation efficiency of fresh agricultural products, blockchain unit variable costs, the level of investment in blockchain technology, and consumer channel preferences on decision-making in dual-channel supply chains for fresh agricultural products.

## 2.2 The use of traceability systems in fresh agricultural products supply chains

Based on the current situation of the promotion of traceability system in fresh agricultural products, some scholars began to study the impact of traceability system on fresh agricultural products supply chain decision making [31–33]. It is found that although traceability systems play a certain role in reducing the distribution loss of fresh agricultural products, traceability systems are based on information systems composed of centralized servers and clients, and it is difficult to eliminate the trust crisis among supply chain members by relying only on a single information node to store, transmit and share information [34]. Meanwhile, due to the lack of transparency of the traceability system, it is still difficult for consumers to obtain complete and true transaction information in the multi-level and complex circulation process of fresh agricultural products, which greatly affects their purchase intention. Therefore, scholars found through further research that, unlike the centralized mechanism of the traceability system, blockchain technology is based on key technologies such as distributed storage, peer-to-peer transmission, and encryption algorithms [35], and has the characteristics of decentralization, information immutability, high security, and high transparency [36]. Bamakan, S. M. H. introduced blockchain technology into a model for evaluating service supply chain performance, addressing ANFIS' reliance on big data and the lack of trust and security in the supply chain [37]. The introduction of blockchain technology into the fresh agricultural products supply chain can effectively solve the problems of low collaboration efficiency, information asymmetry, and high transaction costs upstream and downstream of the supply chain [38, 39]. At the same time, consumers can also quickly trace the origin of the products when they buy the products supported by blockchain technology [40]. Therefore, based on the dual-channel supply chain of fresh agricultural products, this paper analyzes four decision scenarios and gives the optimal strategies under different scenarios by considering blockchain technology and manufacturers' online channel selection.

## 2.3 The impact of blockchain technology on the supply chain sector

Based on the promotion and application of blockchain technology in the supply chain field [41], scholars have mainly discussed the advantages of blockchain technology and its impact on supply chain pricing, finance, channels and other decision-making areas. Chio [42] studied the pricing decision of luxury goods under blockchain technology and concluded that the blockchain system is better than the traditional traceability system. Liang and Xiao [43] based on consumer sensitivity to product authenticity, conducted research focusing only on the fixed costs of blockchain technology and studied the optimal pricing and channel selection in a general product supply chain. Bai et al. [44] constructed a supply chain model with comprehensive consideration of risk avoidance and investment cost, compared and analyzed the changes of supply chain decision before and after the application of blockchain technology, and the coordination contract is constructed to make the supply chain reach coordination. Further, some scholars have introduced blockchain technology into the research related to fresh agricultural products supply chain. Tonnissen and Teuteberg [45] studied the application and impact of blockchain in agricultural products supply chain through practical case studies. Xu et al. [46] found that the implementation of blockchain by manufacturers can increase the greenness of products and also promote supply chain optimization and coordination. Peng et al. [47] found that blockchain-based smart contracts and Digital signature technology can greatly reduce double loss and achieve efficient and high quality circulation in fresh agricultural products supply chain. Feng et al. [48] introduced blockchain-based traceability technology into fresh agricultural products supply chain to guarantee the credibility of agricultural products data through real-time monitoring of multiple nodes. The aforementioned literature mainly considered the impact of introducing blockchain technology on enterprise operation and decision making. Less literature considered the situation of fresh agricultural products supply chain, and mainly studied the improvement and utility of blockchain technology on agricultural products supply chain through case studies [49, 50], without specifically quantifying the economic benefits brought by blockchain technology to improve the circulation time of fresh agricultural products [51]. In this paper, the economic benefits of blockchain technology on improving the distribution time of fresh agricultural products and increasing consumers' trust are quantified by considering two dual-channel structures, namely, direct online sales or online distribution, and the four decision scenarios before and after introducing blockchain technology into the fresh agricultural products supply chain are compared and analyzed.

In summary, scholars have actively explored the decision-making issues in the dual-channel supply chain of fresh agricultural products from various perspectives and achieved certain results. However, there are still gaps in several areas. Firstly, existing research has focused on pricing and coordination issues in the supply chain of fresh agricultural products under single-channel or single dual-channel models. There is a need for additional studies addressing the pricing and channel selection problems in the dual-channel supply chain of fresh agricultural products under different dual-channel structural models. Secondly, most scholars have concentrated on using case studies to explore the impact of blockchain on enterprise operations and decision-making. There is limited research involving the quantification of the economic benefits of blockchain technology in improving the circulation time of fresh agricultural products and increasing consumer trust under a blockchain paradigm. Thirdly, existing literature primarily examines the impact of blockchain technology on the supply chain from a singular perspective, and there is relatively less research that comprehensively considers the impact of consumer preferences, the degree of blockchain usage, and the variable costs of blockchain on simulation results.

## 3 Model description and assumptions

Based on the first two sections, in order to study the three problems raised in the introduction, this section adopts the method of mathematical modeling to transform the studied problems into corresponding mathematical models, and carries out basic assumptions and explanations for the relevant variables in the models.

### 3.1 Model description

In this research, we focus on a single-cycle dual-channel supply chain comprising manufacturers, including farmers, cooperatives, agricultural modernization bases, and leading enterprises, as well as retailers represented by distributors and chain supermarkets. In this context, both manufacturers and retailers aim to maximize their individual interests, resulting in a competitive Stackelberg game between them. Notably, leading enterprises and agricultural industrialization cooperatives hold a dominant position among the manufacturers. Since manufacturers produce a single product, we categorize the supply chain into four models based on the adoption of blockchain technology and the characteristics of online direct sales channels. These models include the following: the non-blockchain technology direct selling mode (NS mode), non-blockchain technology distribution mode (ND mode), blockchain technology direct selling mode (BS mode), and blockchain technology distribution mode (BD mode). The structural diagram is depicted in Fig 1.

NS model: In the dual-channel setup of online direct sales without blockchain technology, fresh agricultural product manufacturers continue to operate through traditional retail channels while also establishing their own online direct sales channels. In this scenario, they are responsible for setting the retail prices for their products in both the traditional retail and direct sales channels, which results in unit direct sales costs during the sales process. Traditional retailers, on the other hand, base their retail prices on the wholesale prices determined by the manufacturers.

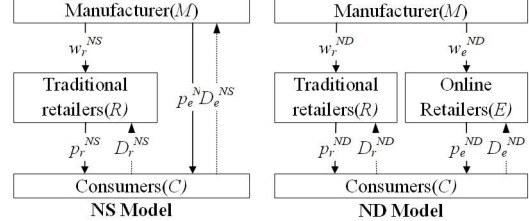

(a) Two dual-channel structures before the adoption of blockchain technology

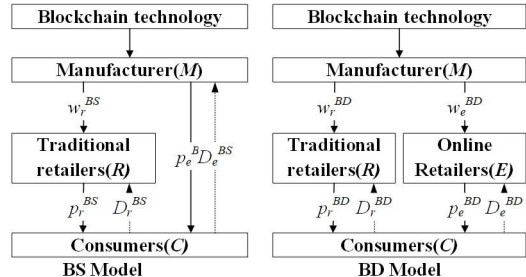

(b) Two dual-channel structures after the adoption of blockchain technology

**Fig 1. Different dual-channel structures before and after the adoption of blockchain technology.**

ND model: In the dual-channel system of online distribution without the use of an alliance chain, fresh agricultural product manufacturers continue to operate through traditional retail channels while also introducing online retail channels. In this setup, fresh agricultural product manufacturers supply their products to both traditional and online retailers at distinct whole-sale prices. Subsequently, traditional retailers and online retailers independently establish their retail prices based on the respective wholesale prices they receive.

BS model: In the dual-channel system of online direct sales utilizing blockchain technology, manufacturers integrate blockchain technology to enhance and revamp their existing enterprise network platform. While still maintaining traditional retail channels, these manufacturers also establish their own online direct sales channels where they set retail prices for their products and directly engage in sales. Because the revamped network platform now includes sales capabilities, there are no unit direct sales costs associated with the sales process. Instead, the traditional unit direct sales cost is converted into maintenance costs for the blockchain system.

BD model: In a dual-channel online distribution system empowered by blockchain technology, manufacturers establish wholesale prices for both offline and online distribution channels. Subsequently, traditional retailers and online retailers independently determine their retail prices based on these wholesale prices and conduct their sales operations accordingly.

## 3.2 Fundamental assumption

Blockchain technology has the capability to gather and share all data throughout the entire production, procurement, processing, transportation, and final sale processes, securely storing this information within blocks. Moreover, the information contained within these blocks can be traced through timestamps, guaranteeing its authenticity and transparency. Utilizing smart contract technology, blockchain can expedite transactions and decrease turnaround times. Building upon these foundational elements, we introduce quantitative analyses based on the concepts of double losses and trust gains associated with circulation time. Table 2. illustrates the configuration and significance of model parameters.

**Assumption 1:** Referring to Liu and Li [51] and Cai et al. [52] method, let the freshness preservation function be $\theta(t) = \theta(0)(1-t^2/T^2)$, and assume that the circulation time of the product is $t_0$ when blockchain technology is not used. Then the freshness of the product when it reaches the consumer is $\theta(t_0)$, and the circulation time $t_0$ drops to $t_1$, when blockchain technology is used. Then the freshness of the product when it reaches the consumer is $\theta(t_1)$, and let the level of the on-chain fresh e-commerce platform put into blockchain technology is $\tau(0 < \tau \leq 1)$, then $t_1 = t_0(1-\tau)$, $0 \leq \theta(t_0) < \theta(t_1) \leq 1$.

**Assumption 2:** Fresh produce loses quantity over time in circulation, with reference to Cai et al. [52] construct an effective output factor function $\varphi(t) = 2 - \exp(\frac{\ln 2}{T}t)$, $\varphi(t) \in [0, 1]$, where exp is a natural constant. To ensure an effective quantity of product, the actual quantity of product that the manufacturer needs to supply to the retailer in the traditional channel is $D_r/\varphi(t)$. Similarly, in the online channels, the actual quantity shipped is $D_e/\varphi(t)$.

**Assumption 3:** Demand for fresh agricultural products in each channel is influenced by the freshness and price of each channel. Given the most obvious characteristics that distinguish fresh agricultural products from ordinary products, it can be argued that the freshness of fresh agricultural products can influence the sales volume of each channel, and that the higher the freshness, the stronger the consumers' willingness to buy at the same price. Referring to Lin

**Table 2. Notation and definitions.**

| Parameters and meaning | Parameters | Significance |
|---|---|---|
| Basic parameters | $\omega$ | Wholesale price per unit of product |
| | $p$ | Sales price per unit of product |
| | $D$ | Market Demand |
| | $t$ | Fresh produce circulation time |
| | $\theta(t)$ | Freshness decay function for fresh agricultural products |
| | $\varphi(t)$ | Effective output factor function for fresh agricultural products |
| | $a$ | Potential market demand capacity |
| | $\pi$ | Profit function |
| Cost parameters | $c_p$ | Unit production cost of fresh agricultural products |
| | $c_s$ | Unit distribution cost of fresh agricultural products |
| | $c_z$ | Unit direct selling cost |
| | $c_\tau$ | Unit variable input cost of applying blockchain |
| Dual channel parameters | $r$ | Traditional channels |
| | $e$ | Online channels |
| | $s$ | Consumer preference for traditional channels |
| | $\beta$ | Channel cross-price elasticity coefficient |
| Blockchain parameters | $k_\tau$ | Blockchain technology cost sensitivity |
| | $\delta$ | The trust gain factor brought by blockchain |
| | $\tau$ | The level of blockchain technology investment |

et al. [53] and Chen [54], the sales volume function for traditional and online channels is:

$$D_r^N = \theta(t_0)(sa - \alpha_1 p_r + \beta_1 p_e) \tag{1}$$

$$D_e^N = \theta(t_0)((1-s)a - \alpha_2 p_e + \beta_2 p_r) \tag{2}$$

where $a$ is the potential market demand, $p_r$ and $p_e$ are the selling prices of the traditional retail channel and the online channel, respectively, $s(0 < s < 1)$ is the degree of consumer preference for the traditional channel, $1\text{-}s(0 < s < 1)$ is the degree of consumer preference for the online channel. This symbol $\alpha_i(i = 1,2)$ denotes the consumer's sensitivity coefficient to the channel price, another symbol $\beta_i(i = 1,2)$ denotes the inter-channel price elasticity coefficient, $0 < \beta_i < \alpha_i \leq 1$, $i = 1,2$. To simplify the calculation of the model, reference is made to Mukhopadhyay [55] and Yan [56], let $\alpha_1 = \alpha_2 = 1$, $\beta_1 = \beta_2 = \beta$.

**Assumption 4:** Unlike conventional centralized information mechanisms, blockchain can gather and store all data in the fresh agricultural products supply chain and create blocks time stamped to track each product from production to final consumption, creating a new trust system with high transparency and security that will boost brand reputation, market demand, and consumer trust. Consequently, assuming that the increase in customer confidence equals [57, 58]. The demand function is currently:

$$D_r^B = \theta(t_1)(sa - p_r + \beta p_e + \delta\tau) \tag{3}$$

$$D_e^B = \theta(t_1)((1-s)a - p_e + \beta p_r + \delta\tau) \tag{4}$$

**Assumption 5:** In general, there is a market demand for dual channels with or without the application of blockchain, and the distribution costs from manufacturer to retailer and the associated costs due to loss of volume are borne by the manufacturer, disregarding other costs, then there is $0 < c_p + c_s < \min(\omega_e, \omega_r) < \min(p_r, p_e)$.

**Assumption 6:** The investment cost of blockchain technology is divided into fixed input cost and variable cost. The fixed input cost is one-time and decreases gradually with the wide-spread application of blockchain technology and the continuous improvement of technology level, so this paper only considers the unit variable cost of blockchain technology $c_\tau = \frac{k_\tau}{2}\tau^2$, $k_\tau$ represents Blockchain technology cost sensitivity [59, 60].

# 4 Dual-channel structure that does not use blockchain technology

Building upon the problem description and decision assumptions outlined in the previous section, we initially addressed the NS and ND models without blockchain technology. This enabled us to determine the optimal pricing strategies and profits for fresh agricultural product manufacturers and retailers within these two models. Subsequently, we draw conclusions and derive pertinent managerial insights through a comparative analysis of the results.

## 4.1 Online direct selling dual-channel model (NS model)

In this model, manufacturers and traditional retailers obey a two-stage Stackelberg game. In the first stage, manufacturers first decide the wholesale price in the traditional $\omega_r^{NS}$ channel and the price for direct online sales $p_e^{NS}$. In the second stage, traditional retailers decide the price of products sold in the traditional channel $p_r^{NS}$.

At this time, the demand of dual-channel is affected by parameters such as freshness of fresh agricultural products, cross-price sensitivity coefficient, and channel preference of consumers. The function is shown as follows:

$$D_r^{NS} = \theta(t_0)(sa - p_r^{NS} + \beta p_e^{NS}) \tag{5}$$

$$D_e^{NS} = \theta(t_0)((1-s)a - p_e^{NS} + \beta p_r^{NS}) \tag{6}$$

The profit function of each member of the dual-channel supply chain can be formulated as:

$$\pi_M^{NS} = (\omega_r^{NS} - \frac{c_p + c_s}{\varphi(t_0)})D_r^{NS} + (p_e^{NS} - c_z - \frac{c_p + c_s}{\varphi(t_0)})D_e^{NS} \tag{7}$$

$$\pi_R^{NS} = (p_r^{NS} - \omega_r^{NS})D_r^{NS} \tag{8}$$

The inverse induction method is used to bring (5) and (6) into (7), and find the second partial derivative with respect to $p_r^{NS}$. Then $\frac{\partial^2 \pi_R^{NS}}{\partial p_r^{NS2}} = -2\theta(t_0) < 0$, so $\pi_R^{NS}$ is a strictly concave function about $p_r^{NS}$. As this time, let $\frac{\partial \pi_R^{NS}}{\partial p_r^{NS}} = 0$, solve $p_r^{NS}$ and take (6) the derivative of $\omega_r^{NS}$ and $p_e^{NS}$ to obtain the optimal wholesale price $\omega_r^{NS}$ and the optimal online direct selling retail price $p_e^{NS}$, thus obtaining $p_r^{NS}$. Substitute the optimal solution into the profit function to obtain the optimal profit $\pi_R^{NS}$ and $\pi_M^{NS}$. The optimal equilibrium solution is shown in Table 3.

**Table 3. Equilibrium solutions in NS and ND models.**

|  | NS Model | ND Model |
|---|---|---|
| $p_e^*$ | $\frac{a-sa+\beta sa}{2(1-\beta^2)} + \frac{c_z}{2} + A$ | $\frac{3(2-\beta^2)(1-s)a+(5-2\beta^2)s\beta a}{2(1-\beta^2)(4-\beta^2)} + \frac{A}{2-\beta}$ |
| $p_r^*$ | $\frac{(3-\beta^2)sa+2(1-s)\beta a}{4(1-\beta^2)} + \frac{(1+\beta)A}{2} + \frac{\beta c_z}{4}$ | $\frac{3(2-\beta^2)sa+(5-2\beta^2)(1-s)\beta a}{2(1-\beta^2)(4-\beta^2)} + \frac{A}{2-\beta}$ |
| $\omega_e^*$ | —— | $\frac{a-sa+s\beta a}{2(1-\beta^2)} + A$ |
| $\omega_r^*$ | $\frac{sa+\beta a-\beta sa}{2(1-\beta^2)} + A$ | $\frac{sa+\beta a-s\beta a}{2(1-\beta^2)} + A$ |

## 4.2 Online distribution dual-channel model (ND model)

In this model, manufacturers and traditional retailers obey a two-stage Stackelberg game. In the first stage, manufacturers first determine the wholesale price in the traditional channel $\omega_r^{ND}$ and determine the wholesale price in the online channel $\omega_e^{ND}$. In the second stage, the traditional retailers and online retailers, respectively, to determine the product retail price in traditional channels and online channels $p_r^{ND}$ and $p_e^{ND}$.

Similar 4.1, the demand functions for the traditional and online channels can be expressed by:

$$D_r^{ND} = \theta(t_0)(sa - p_r^{ND} + \beta p_e^{ND}) \tag{9}$$

$$D_e^{ND} = \theta(t_0)((1-s)a - p_e^{ND} + \beta p_r^{ND}) \tag{10}$$

The profit function of each member of the dual-channel supply chain can be formulated as:

$$\pi_M^{ND} = \omega_r^{ND}D_r^{ND} + \omega_e^{ND}D_e^{ND} - (c_p + c_s)(D_r^{ND} + D_e^{ND})/\varphi(t_0) \tag{11}$$

$$\pi_R^{ND} = (p_r^{ND} - \omega_r^{ND})D_r^{ND} \tag{12}$$

$$\pi_E^{ND} = (p_e^{ND} - \omega_e^{ND})D_e^{ND} \tag{13}$$

The inverse induction method is used to bring (9) and (10) into (12) and (13), and the first-order partial conductance parallel vertical solution is obtained for $p_r^{ND}$ and $p_e^{ND}$. The results are brought into (11), and the first-order partial conductance parallel vertical solution is obtained for $\omega_r^{ND}$ and $\omega_e^{ND}$. The optimal equilibrium solution is shown in Table 3.

Where $A = \frac{c_p + c_s}{2\varphi(t_0)}$.

## 4.3 Comparative analysis of NS model and ND model

**Proposition 1** When not using blockchain technology:

a. $\frac{\partial \omega_r^{I*}}{\partial \varphi(t_0)} < 0, \frac{\partial \omega_e^{ND*}}{\partial \varphi(t_0)} < 0, \frac{\partial p_r^{I*}}{\partial \varphi(t_0)} < 0, \frac{\partial P_e^{I*}}{\partial \varphi(t_0)} < 0.$

b. $\frac{\partial D_r^{I*}}{\partial \varphi(t_0)} > 0, \frac{\partial D_e^{I*}}{\partial \varphi(t_0)} > 0, \frac{\partial \pi_R^{I*}}{\partial \varphi(t_0)} > 0, \frac{\partial \pi_M^{I*}}{\partial \varphi(t_0)} > 0, \frac{\partial \pi_E^{ND*}}{\partial \varphi(t_0)} > 0.$

c. $\frac{\partial \omega_r^{I*}}{\partial \theta(t_0)} = 0, \frac{\partial \omega_e^{ND*}}{\partial \theta(t_0)} = 0, \frac{\partial p_r^{I*}}{\partial \theta(t_0)} = 0, \frac{\partial p_e^{I*}}{\partial \theta(t_0)} = 0.$

d. $\frac{\partial D_r^{I*}}{\partial \theta(t_0)} > 0, \frac{\partial D_e^{I*}}{\partial \theta(t_0)} > 0, \frac{\partial \pi_R^{I*}}{\partial \theta(t_0)} > 0, \frac{\partial \pi_M^{I*}}{\partial \theta(t_0)} > 0, \frac{\partial \pi_E^{ND*}}{\partial \theta(t_0)} > 0.$

Where $I$ = NS, ND.

Proof of Proposition 1. See S1 Appendix.

Proposition 1 indicates that whether fresh product manufacturers adopt online direct sales or online distribution, both dual-channel wholesale prices and selling prices are negatively correlated with the effective output ratio $\varphi(t_0)$. Meanwhile, dual-channel demand and the profits of supply chain members are influenced by double losses, positively correlated with the effective output ratio $\varphi(t_0)$ and freshness $\theta(t_0)$. Given the perishable nature of fresh agricultural products, as the circulation time increases, the losses and reduced freshness of fresh agricultural products become more severe, resulting in lower effective output. Therefore, fresh product manufacturers can only reduce the cost increase caused by double losses by increasing wholesale and online direct sales prices. Additionally, traditional retailers raise their selling

prices to ensure their own profits. In the end, this leads to a decrease in dual-channel market demand and the profits of supply chain members. The more severe the losses, the lower the corresponding profit levels.

**Proposition 2** When not using blockchain technology:

a. $\frac{\partial p_e^{I*}}{\partial s} < 0, \frac{\partial p_r^{I*}}{\partial s} > 0, \frac{\partial \omega_r^{I*}}{\partial s} > 0, \frac{\partial \omega_e^{ND*}}{\partial s} < 0.$

b. When $0 < s \leq 1/2$, $\omega_e^{ND*} \geq \omega_r^{ND*}$; and when $1/2 < s < 1$, $\omega_e^{ND*} < \omega_r^{ND*}$.

c. $\frac{\partial p_e^{NS*}}{\partial c_z} > 0, \frac{\partial p_r^{NS*}}{\partial c_z} > 0, \frac{\partial \omega_r^{NS*}}{\partial c_z} = 0.$

Proof of Proposition 2. See S1 Appendix.

Proposition 2 indicates that in the absence of blockchain technology, whether in the online direct sales or online distribution model, the wholesale prices and selling prices in traditional channels increase as consumer preferences for traditional channels increase. Conversely, wholesale prices and selling prices in online channels decrease as consumer preferences for online channels increase. In the online distribution dual-channel model, when consumers prefer online channels, fresh product manufacturers set wholesale prices for online retailers no lower than those for traditional retailers, and vice versa when consumers prefer traditional channels. In the online direct sales dual-channel model, the wholesale price decisions of fresh product manufacturers are not influenced by online direct sales costs, but the selling prices in both channels increase as online direct sales costs increase. This is because when consumer preferences shift towards traditional channels, online channels lose their market advantage. Consequently, fresh product manufacturers and online retailers can only attract consumers by lowering prices. In the online direct sales model, as direct sales costs increase, fresh product manufacturers shift some of the costs to consumers by raising online direct sales prices. At the same time, to maximize their own interests, fresh product manufacturers should keep wholesale prices for traditional retailers consistent to ensure normal revenue in the traditional channel. Considering their cooperative relationship, traditional retailers also raise the selling prices in their traditional channels to mitigate price competition between the two channels.

**Proposition 3** When not using blockchain technology:

a. $\frac{\partial \pi_R^{I*}}{\partial s} > 0, \frac{\partial \pi_E^{ND*}}{\partial s} < 0.$

b. When $c_z \geq \frac{\varphi(t_0)(2-3s+s\beta)a}{(1+\beta)(2-\beta)} - \frac{(1-\beta)(c_p+c_s)}{2-\beta}, \frac{\partial \pi_M^{NS*}}{\partial s} \leq 0$; and vice versa $\frac{\partial \pi_M^{NS*}}{\partial s} > 0.$

c. When $0 < s \leq 1/2, \frac{\partial \pi_M^{ND*}}{\partial s} \leq 0$; and vice versa $\frac{\partial \pi_M^{ND*}}{\partial s} > 0.$

Proof of Proposition 3. See S1 Appendix.

Proposition 3 reveals that consumer channel preferences affect the pricing of dual-channel products, thereby influencing the profit levels of manufacturers and retailers in the dual-channel system. The following conclusions can be drawn:

1. In the absence of blockchain technology, the profit of traditional retailers increases as consumer preferences for traditional channels increase, while the profit of online retailers consistently decreases. Specifically, the profit dynamics for fresh product manufacturers differ between the two models. In the online direct sales model, when the manufacturer's online direct sales costs exceed a certain threshold, manufacturer profits decrease as consumer preferences shift towards traditional channels. In the online distribution model, the profit of fresh product manufacturers exhibits a "U"-shaped pattern as consumer preferences for

traditional channels increase, reaching its lowest point at $s = 1/2$. With increasing consumer preferences for traditional channels, traditional channels gain a competitive advantage.

2. In the online direct sales model, the wholesale price in traditional channels is not affected by direct sales costs. Therefore, when online direct sales costs become excessively high, the profit gained by fresh product manufacturers in the online direct sales channel is insufficient to cover their direct sales costs, leading to a decline in manufacturer profits. In the online distribution model, as Proposition 2 suggests, as consumer preferences for traditional channels increase, fresh product manufacturers increase wholesale prices for traditional retailers to gain profit benefits and may appropriately lower wholesale prices for online retailers. When $s < 1/2$, as consumer preferences for traditional channels increase, the profit loss incurred by manufacturers from lowering wholesale prices for online retailers is greater than the profit gain from raising wholesale prices for traditional retailers, resulting in a decrease in manufacturer profits. However, when $s > 1/2$, as consumer preferences for traditional channels increase, the profit loss from lowering wholesale prices for online retailers is smaller than the profit gain from raising wholesale prices for traditional retailers, leading to an increase in manufacturer profits. Ultimately, with changes in $S$, this relationship follows a "U"-shaped pattern.

**Proposition 4** When not using blockchain technology:

a. When $0 < c_z \leq \frac{2(1-s)a+s\beta a}{4-\beta^2} - \frac{2A(1-\beta)}{2-\beta}$, $p_e^{NS*} \leq p_e^{ND*}$, $p_r^{NS*} \leq p_r^{ND*}$; and vice versa $p_e^{NS*} > p_e^{ND*}$, $p_r^{NS*} > p_r^{ND*}$.

b. When $0 < c_z \leq \frac{2(1-s)a+s\beta a}{4-\beta^2} - \frac{2A(1-\beta)}{2-\beta}$, $\pi_R^{NS*} \leq \pi_R^{ND*}$; and vice versa $\pi_R^{NS*} > \pi_R^{ND*}$.

c. When $0 < c_z \leq \frac{(4-\beta^2+\sqrt{8-2\beta^2})(s\beta a+2(1-s)a)}{(2-\beta^2)(4-\beta^2)} - \frac{2A(2-\beta-\beta^2)}{(2-\beta^2)(4-\beta^2)}$, $\pi_M^{NS*} \geq \pi_M^{ND*}$; and vice versa, $\pi_M^{NS*} < \pi_M^{ND*}$.

Proof of Proposition 4. See S1 Appendix.

Proposition 4 suggests that in the absence of blockchain adoption in the supply chain, the pricing and profits of fresh agricultural products in two dual-channel models are influenced by the manufacturer's direct selling costs $c_z$. When the unit direct selling cost is lower than a certain threshold, both the channel prices and profits of traditional retailers in the network direct sales model are lower than the network distribution model, while the profits of fresh manufacturers are higher than the network distribution model. In the network direct sales model without the use of blockchain technology, fresh manufacturers incur direct selling costs. When these direct selling costs are low, fresh manufacturers sell through network channels at prices lower than the network distribution model, leading to an increase in network channel demand and profits. However, for traditional retailers, Proposition 2 shows that wholesale prices in both models are consistent. When direct selling costs are low, manufacturers can leverage their direct selling advantage to sell through network channels at lower prices. Therefore, traditional retailers, in order to maximize their own interests, also need to implement price reduction measures to increase their channel competitiveness. Given that fresh agricultural product manufacturers have a significant pricing advantage in network direct sales, the price reductions by traditional retailers are not sufficient to attract consumers, resulting in a decrease in demand for traditional channels and a decrease in the profits of traditional retailers.

## 5 Dual-channel structure using blockchain technology

In this section, we commence by determining the optimal decisions within the online direct selling model (BS model) and online distribution model (BD model) that incorporate blockchain technology into the supply chain system. We conduct a comparative analysis, contrasting the wholesale prices, direct selling prices, retail prices, retailer profits, and manufacturer profits between model BS and model BD. Subsequently, we derive conclusions and formulate corresponding propositions based on our findings.

### 5.1 Online direct selling dual-channel model (BS model)

When applying blockchain technology, the circulation time of fresh agricultural products is reduced from $t_0$ to $t_1$. The decision order is the same as the NS model in Section 4.1, so traditional and online channels' demand functions are:

$$D_r^{BS} = \theta(t_1)(sa - p_r^{BS} + \beta p_e^{BS} + \delta\tau) \tag{14}$$

$$D_e^{BS} = \theta(t_1)((1-s)a - p_e^{BS} + \beta p_r^{BS} + \delta\tau) \tag{15}$$

The profit function of each member of the dual-channel supply chain can be formulated as:

$$\pi_M^{BS} = (\omega_r^{BS} - c_\tau)D_r^{BS} + (p_e^{BS} - c_\tau)D_e^{BS} - (c_p + c_s)(D_r^{BS} + D_e^{BS})/\varphi(t_1) \tag{16}$$

$$\pi_R^{BS} = (p_r^{BS} - \omega_r^{BS})D_r^{BS} \tag{17}$$

The inverse induction method is used to bring (14) and (15) into (17), take the derivative of $p_r^{BS}$ and solve it. The results are brought into (16), and the first-order partial conductance parallel vertical solution is obtained for $\omega_r^{BS}$ and $p_e^{BS}$ to obtain the optimal whole price $\omega_r^{BS}$, the optimal online direct selling retail price $p_e^{BS}$, and the optimal traditional retail price $p_r^{BS}$. By bringing the optimal solution into the profit function, the optimal profit are obtained. The optimal equilibrium solution is shown in Table 4.

### 5.2 Online distribution dual-channel model (BD model)

The decision order is the same as the ND model in Section 4.2, so the demand functions for the traditional and online channels can be expressed by:

$$D_r^{BD} = \theta(t_1)(sa - p_r^{BD} + \beta p_e^{BD} + \delta\tau) \tag{18}$$

$$D_e^{BD} = \theta(t_1)((1-s)a - p_e^{BD} + \beta p_r^{BD} + \delta\tau) \tag{19}$$

**Table 4. Equilibrium solutions in BS and BD models.**

| | BS Model | BD Model |
|---|---|---|
| $p_e^*$ | $\frac{(1-s+\beta s)a+(1+\beta)\delta\tau}{2(1-\beta^2)} + B$ | $\frac{3(2-\beta^2)(1-s)a+(5-2\beta^2)\beta sa}{2(1-\beta^2)(4-\beta^2)} + \frac{B}{(2-\beta)} + \frac{(6+5\beta-3\beta^2-2\beta^3)\delta\tau}{2(1-\beta^2)(4-\beta^2)}$ |
| $p_r^*$ | $\frac{((3-\beta^2)s+2\beta(1-s))a}{4(1-\beta^2)} + \frac{(1+\beta)B}{2} + \frac{(3+2\beta-\beta^2)\delta\tau}{4(1-\beta^2)}$ | $\frac{3(2-\beta^2)sa+(5-2\beta^2)(1-s)\beta a}{2(1-\beta^2)(4-\beta^2)} + \frac{B}{2-\beta} + \frac{(6+5\beta-3\beta^2-2\beta^3)\delta\tau}{2(1-\beta^2)(4-\beta^2)}$ |
| $\omega_e^*$ | —— | $\frac{(1-s)a+\beta sa+(1+\beta)\delta\tau}{2(1-\beta^2)} + B$ |
| $\omega_r^*$ | $\frac{(s+\beta(1-s))a+(1+\beta)\delta\tau}{2(1-\beta^2)} + B$ | $\frac{sa+(1-s)\beta a+(1+\beta)\delta\tau}{2(1-\beta^2)} + B$ |

The profit function of each member of the dual-channel supply chain can be formulated as:

$$\pi_M^{BD} = (\omega_r^{BD} - c_\tau)D_r^{BD} + (\omega_e^{BD} - c_\tau)D_e^{BD} - (c_p + c_s)(D_r^{BD} + D_e^{BD})/\varphi(t_1) \tag{20}$$

$$\pi_R^{BD} = (p_r^{BD} - \omega_r^{BD})D_r^{BD} \tag{21}$$

$$\pi_E^{BD} = (p_e^{BD} - \omega_e^{BD})D_e^{BD} \tag{22}$$

The inverse induction method is used to bring (18) and (19) into (21) and (22), and the first-order partial conductance parallel vertical solution is obtained for $p_r^{BD}$ and $p_e^{BD}$. The results are brought into (20), and the first-order partial conductance parallel vertical solution is obtained for $\omega_r^{BD}$ and $\omega_e^{BD}$. The optimal equilibrium solution is shown in Table 4.

Where $B = \frac{1}{2}\left(\frac{c_p + c_s}{\varphi(t_1)} + c_\tau\right)$.

## 5.3 Comparative analysis of BS model and BD model

**Proposition 5** When adopting blockchain:

a. $\frac{\partial^2 p_e^{J*}}{\partial \tau^2} > 0, \frac{\partial^2 p_r^{J*}}{\partial \tau^2} > 0, \frac{\partial^2 \omega_r^{J*}}{\partial \tau^2} > 0, \frac{\partial^2 \omega_e^{BD*}}{\partial \tau^2} > 0$.

b. $\frac{\partial^2 D_e^{J*}}{\partial \tau^2} < 0, \frac{\partial^2 D_r^{J*}}{\partial \tau^2} < 0, \frac{\partial^2 \pi_R^{J*}}{\partial \tau^2} < 0, \frac{\partial^2 \pi_M^{J*}}{\partial \tau^2} < 0, \frac{\partial^2 \pi_E^{BD*}}{\partial \tau^2} < 0$.

Where $J$ = BS, BD.

Proof of Proposition 5. See S1 Appendix.

Proposition 5 indicates that with the introduction of blockchain technology, the effective output $\varphi(t_1)$ and freshness $\theta(t_1)$ of fresh agricultural products during transportation are both improved. In the scenario where blockchain is applied, the sales prices and wholesale prices in both models show a "U"-shaped variation as $\tau$ (the level of blockchain technology investment) increases. The impacts on channel demand and supply chain member profits are influenced by the combined effects of blockchain investment benefits and costs, primarily manifested in three aspects: the gains from reducing quantity loss and freshness loss, the gains from increased consumer trust, and the costs incurred from technological investments (these three aspects represent the rate of change with respect to the degree of blockchain implementation). Dual-channel demand and profits exhibit an inverted "U"-shaped variation as the degree of blockchain implementation increases, indicating the existence of a peak point in supply chain profits. This can serve as the optimal investment point for blockchain technology.

**Proposition 6** When adopting blockchain:

a. $\frac{\partial p_e^{J*}}{\partial s} < 0, \frac{\partial p_r^{J*}}{\partial s} > 0, \frac{\partial \omega_r^{J*}}{\partial s} > 0, \frac{\partial \omega_e^{BD*}}{\partial s} < 0$.

b. When $0 < s \le 1/2$, $p_e^{BS*} \ge w_r^{BS*}$, $\omega_e^{BD*} \ge \omega_r^{BD*}$; and vice versa $p_e^{BS*} < \omega_r^{BS*}$, $\omega_e^{BD*} < \omega_r^{BD*}$.

Proof of Proposition 6. See S1 Appendix.

Proposition 6 shows that when blockchain is adopted, the relationship between the pricing decision of the dual-channel fresh produce supply chain and the degree of consumer preference for traditional channels is consistent with the conclusion of Proposition 2.

**Proposition 7** When adopting blockchain:

a. $\omega_r^{BD*} = \omega_r^{BS*}, p_e^{BS*} < p_e^{BD*}, p_r^{BS*} < p_r^{BD*}$.

b. $D_r^{BS*} < D_r^{BD*}, \pi_R^{BS*} < \pi_R^{BD*}, D_e^{BS*} > D_e^{BD*}, \pi_M^{BS*} > \pi_M^{BD*}$.

Proof of Proposition 7. See S1 Appendix.

Proposition 7 demonstrates that when adopting blockchain technology, both the direct sales model and the distribution model have the same wholesale prices. In the distribution model with the use of blockchain technology, the manufacturer's wholesale prices for traditional retailers and online retailers are also the same. In comparison to the distribution model, the online retail prices, traditional channel retail prices, traditional channel sales prices, traditional channel demand, and profits for traditional retailers are all lower in the direct sales model, while network channel demand and profits for fresh agricultural product manufacturers are higher. This is because in the context of blockchain technology adoption, regardless of the dual-channel structure model chosen by fresh agricultural product manufacturers, traditional offline channels are an indispensable part of the channel composition. Therefore, for fresh manufacturers to maximize their own interests, they need to keep the wholesale prices for traditional retailers consistent to ensure the normal cooperation of traditional channels. In contrast to the network distribution model, in the case of blockchain application, manufacturers save on direct selling costs. Consequently, manufacturers can offer lower direct selling prices. At the same time, traditional retailers are forced to implement price reduction measures. As a result, manufacturers have higher profits in the direct sales model (BS model), while traditional retailers' profits are higher in the distribution model (BD model).

# 6 Analysis of the impact of blockchain technology on dual-channel supply chain decision making for fresh agricultural products

In this section, we begin by performing a sensitivity analysis of pertinent parameters. Subsequently, we assess the influence of blockchain technology on the channel selection strategies of fresh agricultural product manufacturers. Finally, we delve into the blockchain investment decisions made by fresh agricultural product manufacturers under various channel choices.

## 6.1 Comparative analysis of NS model and BS model

**Proposition 8** In the online direct sales dual-channels:

a. When $c_\tau > C - \frac{\delta\tau}{2(1-\beta)}$ there is $\omega_r^{BS*} > \omega_r^{NS*}$; and when $0 < c_\tau \leq C - \frac{\delta\tau}{2(1-\beta)}$, there is $\omega_r^{BS*} \leq \omega_r^{NS*}$.

b. When $c_\tau > C + \frac{\beta c_z}{1+\beta} - \frac{(3-\beta)\delta\tau}{1-\beta}$, there is $p_r^{BS*} > p_r^{NS*}$; and when $0 < c_\tau \leq C + \frac{\beta c_z}{1+\beta} - \frac{(3-\beta)\delta\tau}{1-\beta}$, there is $p_r^{BS*} \leq p_r^{NS*}$.

c. When $c_\tau > C - \frac{\delta\tau}{1-\beta} + c_z$, there is $p_e^{BS*} > p_e^{NS*}$; and when $0 < c_\tau \leq C - \frac{\delta\tau}{1-\beta} + c_z$, there is $p_e^{BS*} \leq p_e^{NS*}$.

Where $C = \frac{c_p+c_s}{\varphi(t_0)} - \frac{c_p+c_s}{\varphi(t_1)}$.

Proof of Proposition 8. See S1 Appendix.

Proposition 8 indicates that in the network direct sales model with the application of blockchain technology, if the unit variable cost of blockchain technology is low, the wholesale price in the traditional channel will be lower than when blockchain is not adopted. If the unit variable cost of blockchain technology is low and satisfies a certain relationship with unit direct selling costs, the increase in consumer trust brought by blockchain technology, and the improvement in the effective output ratio, both the retail price in the traditional channel and the direct selling price in the network channel will be lower than when blockchain is not adopted. This is because, after adopting blockchain technology, on one hand, manufacturers can save on direct selling costs, and with the improvement in the effective output ratio, the production costs for manufacturers decrease. Therefore, fresh agricultural product

manufacturers will reduce their selling prices and wholesale prices, further stimulating consumers and increasing actual market demand. On the other hand, traditional retailers, while benefiting from the manufacturer's investment in blockchain, will also lower the retail prices in the traditional channel to compete effectively.

**Proposition 9** In the online direct sales dual-channels:

a. When $0 < c_\tau \leq c_\tau^{RS*}$, there is $\pi_R^{BS*} \geq \pi_R^{NS*}$; When $c_\tau > c_\tau^{RS*}$, there is $\pi_R^{BS*} < \pi_R^{NS*}$.

b. When $0 < c_\tau \leq c_\tau^{MS*}$, there is $\pi_M^{BS*} \geq \pi_M^{NS*}$; When $c_\tau > c_\tau^{MS*}$, there is $\pi_M^{BS*} < \pi_M^{NS*}$.

Proof of Proposition 9. See S1 Appendix.

Proposition 9 indicates that in the online direct sales model, if the per-unit blockchain technology transformation cost is lower than a certain threshold and there exists a specific relationship between the per-unit direct sales cost, the increase in consumer trust resulting from blockchain technology, and the improvement in the effective output ratio, then the profits of fresh product manufacturers and traditional retailers are higher when using blockchain technology compared to when not using it. Conversely, if the per-unit blockchain technology transformation cost is higher, then profits decrease. This is because when the per-unit transformation cost of blockchain technology is relatively high, fresh product manufacturers and traditional retailers may pass on some of the costs to consumers by raising prices. This, in turn, leads to a decrease in consumer demand. Since the positive effect of higher prices on per-unit net benefits is smaller than the negative effect of reduced demand, both parties experience a decrease in profits.

## 6.2 Comparative analysis of ND model and BD model

**Proposition 10** In the online distribution dual-channels:

a. When $c_\tau > C - \frac{\delta\tau}{1-\beta}$, there are $\omega_r^{BD*} > \omega_r^{ND*}, \omega_e^{BD*} > \omega_e^{ND*}$; and when $0 < c_\tau \leq C - \frac{\delta\tau}{1-\beta}$, there are $\omega_r^{BD*} \leq \omega_r^{ND*}, \omega_e^{BD*} \leq \omega_e^{ND*}$.

b. When $c_\tau > C - \frac{(6-\beta-2\beta^2)\delta\tau}{(1-\beta)(2+\beta)}$, there are $p_r^{BD*} > p_r^{ND*}, p_e^{BD*} > p_e^{ND*}$; and when

$0 < c_\tau \leq C - \frac{(6-\beta-2\beta^2)\delta\tau}{(1-\beta)(2+\beta)}$, there are $p_r^{BD*} \leq p_r^{ND*}, p_e^{BD*} \leq p_e^{ND*}$.

Proof of Proposition 10. See S1 Appendix.

Proposition 10 indicates that in the online distribution model, if the per-unit blockchain technology transformation cost exceeds a certain threshold and there exists a specific relationship between the increase in consumer trust and the improvement in the effective output ratio resulting from blockchain technology, the equilibrium wholesale price after adopting blockchain technology is higher than when not using blockchain. This aligns with the conclusions drawn under the direct sales model. With the adoption of blockchain technology, the circulation time of fresh agricultural products is shortened, the effective output ratio is increased, product authenticity is ensured, and consumers are willing to pay higher costs. Therefore, both traditional retailers and online retailers raise their retail prices.

**Proposition 11** In the online distribution dual-channels:

a. When $0 < c_\tau \leq c_\tau^{RD*}$, $\pi_R^{BD*} \geq \pi_R^{ND*}$; when $c_\tau > c_\tau^{RD*}$, $\pi_R^{BD*} < \pi_R^{ND*}$.

b. When $0 < c_\tau \leq c_\tau^{ED*}$, $\pi_E^{BD*} \geq \pi_E^{ND*}$; when $c_\tau > c_\tau^{ED*}$, $\pi_E^{BD*} < \pi_E^{ND*}$.

c. When $0 < c_\tau \leq c_\tau^{MD*}$, $\pi_M^{BD*} \geq \pi_M^{ND*}$; When $c_\tau > c_\tau^{MD*}$, $\pi_M^{BD*} < \pi_M^{ND*}$.

Proof of Proposition 11. See S1 Appendix.

Proposition 11 suggests that, in the online distribution model, the profits of supply chain members are influenced by the blockchain transformation costs. When the blockchain transformation costs are less than a certain threshold, the profits of fresh agricultural product manufacturers, traditional retailers, and online retailers are higher than when not using blockchain. In actual decision-making, controlling the blockchain transformation costs within a certain threshold is beneficial to all supply chain members when adopting blockchain. This is because, despite a slight increase in wholesale prices when adopting blockchain technology, both traditional retailers and online retailers raise their retail prices. This leads to an increase in consumer trust, an increase in customer demand, and higher profits for traditional retailers and online retailers compared to not using blockchain technology. When the costs associated with investing in blockchain technology are kept within a certain range, it is possible to achieve an overall increase in supply chain profits when adopting blockchain technology.

## 7 Numerical analysis

In the previous sections, we conducted theoretical analyses to examine the influence of the level of blockchain technology investment, blockchain variable costs, and consumer channel preferences on pricing decisions and profit levels within a dual-channel supply chain. In this section, we will utilize numerical simulations to sequentially assess how variable costs associated with blockchain technology, unit direct selling costs in the direct sales channel, and consumer channel preferences impact the decisions made by supply chain members.

In this section, referring to the literature of Li and Zhao [34], we take as an example a leading enterprise in Yantai, Shandong Province, China, which produces high-quality cherries, supplies a traditional fruit supermarket in Beijing, and conducts e-direct sales on the Tmall Supermarket platform. After field investigation and data sorting, cherries have a unit production cost $c_p = 8$ thousand yuan/ton, ten thousand yuan, a unit transportation cost $c_s = 12$ thousand yuan/ton, and a potential market demand in the Beijing urban area $a = 10$ tons, the cross-price elasticity coefficient is a = 4, and the manufacturer's unit cost for online direct sales is $\beta = 0.4$, and the manufacturer's unit cost for online direct sales is $c_z = 6$ thousand yuan/ton, the trust gain coefficient brought by blockchain is δ = 0.4, cherries undergo cold-chain transportation from harvest to consumers and retailers in the Beijing area, taking $t_0 = 4$ days, with a lifecycle of $T = 10$ days. Based on assumptions 1 and 2, without the use of blockchain technology, the effective output ratio $\varphi(t_0) = 2 - \exp^{\ln 2(t_0/T)} = 0.68$ and the freshness $\theta(t_0) = 1 - t_0^2/T^2 = 0.84$. Considering the sensitivity of cherry freshness to circulation efficiency, to reduce product losses and improve freshness, blockchain technology is proposed. With blockchain, the product circulation time is shortened to $t_1 = 2$ days, the effective output ratio increases to $\varphi(t_1) = 2 - \exp^{\ln 2(t_1/T)} = 0.85$, and the freshness improves to $\theta(t_1) = 1 - t_1^2/T^2 = 0.96$. The optimal pricing under the influence of relevant parameters and the optimal dual-channel sales model for supply chain members are illustrated in the following diagrams.

Propositions 2, 3, and 6 demonstrate that consumer channel preferences impact market demand and indirectly alter the wholesale and retail prices of fresh agricultural products, consequently affecting the profit levels of supply chain members. From Fig 2, we can observe that, regardless of whether blockchain is adopted and when $c_\tau = 1$, the profit of traditional retailers is positively correlated with the degree of consumer preference for traditional channels $S$. The profit of online retailers is negatively correlated with the degree of consumer preference for traditional channels $S$. The profit of fresh agricultural product manufacturers initially decreases and then increases with the degree of consumer preference for traditional channels $S$. Furthermore, from Fig 2(B) and 2(D), we can infer that in the distribution model, whether

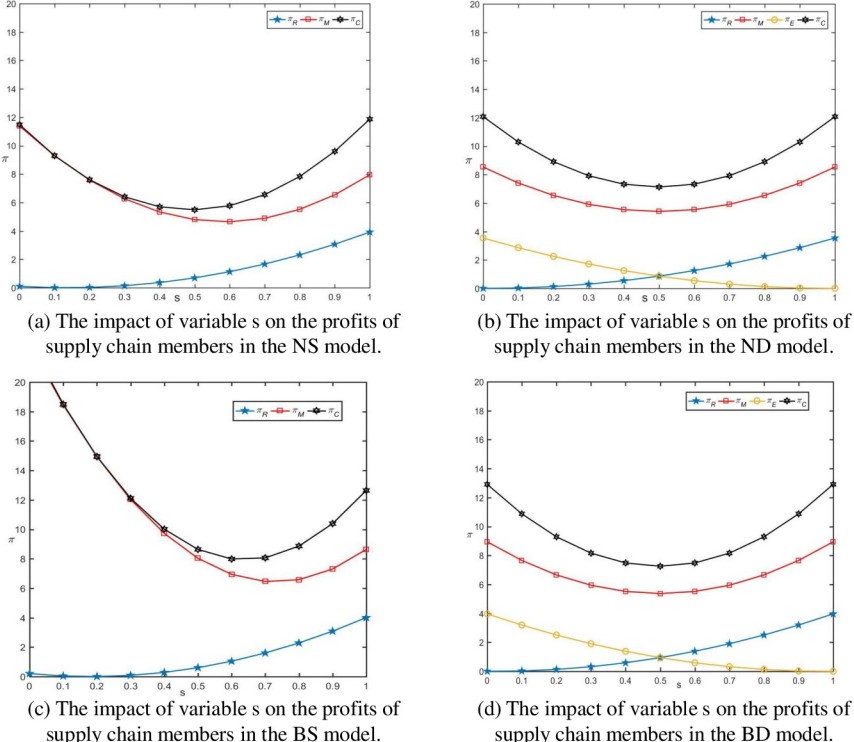

(a) The impact of variable s on the profits of supply chain members in the NS model.

(b) The impact of variable s on the profits of supply chain members in the ND model.

(c) The impact of variable s on the profits of supply chain members in the BS model.

(d) The impact of variable s on the profits of supply chain members in the BD model.

**Fig 2. The impact of variable *s* on the profits of supply chain members.**

or not blockchain is used, the price differences between channels stimulate consumer demand to a certain extent. This increase in demand, coupled with higher channel profits compared to losses, results in an overall supply chain profit that exhibits a "U"-shaped variation with the degree of consumer preference for traditional channels *S*. The supply chain profit reaches its lowest point when *s* = 1/2. These observations highlight the complex interplay between consumer preferences, pricing, and profitability in the context of a dual-channel supply chain for fresh agricultural products.

As indicated by Proposition 5, the impact of the level of blockchain technology investment on supply chain member profits depends on the coupling effect between the gains from reducing double losses and increasing trust and the costs of technology investment. These gains and costs are each represented by linear and quadratic functions with respect to technology investment. When adopting blockchain and taking *S* = 0.45, the results are shown in Figs 3 and 4, respectively. With the increase in the level of technology investment $\tau$, supply chain profits first increase and then decrease, exhibiting an inverted "U" shape, indicating the existence of a peak point for supply chain profits. In the initial stages of blockchain technology investment, the level of reducing double losses and increasing trust is not very significant. In the middle stages of technology investment, as the level of technology investment $\tau$ increases, the gains outweigh the costs, leading to a significant improvement in supply chain profits. In the later stages of technology investment, due to the increasing difficulty in improving the technology, it reaches a bottleneck, resulting in diminishing returns from technology investment.

The cost coefficient $k_\tau$ of blockchain technology represents the difficulty of technology investment, which affects the variable investment cost $c_\tau$. An increase in $k_\tau$ will result in an increase in variable costs under the same blockchain technology benefits, leading to a reduction in profit levels, as shown in Fig 4.

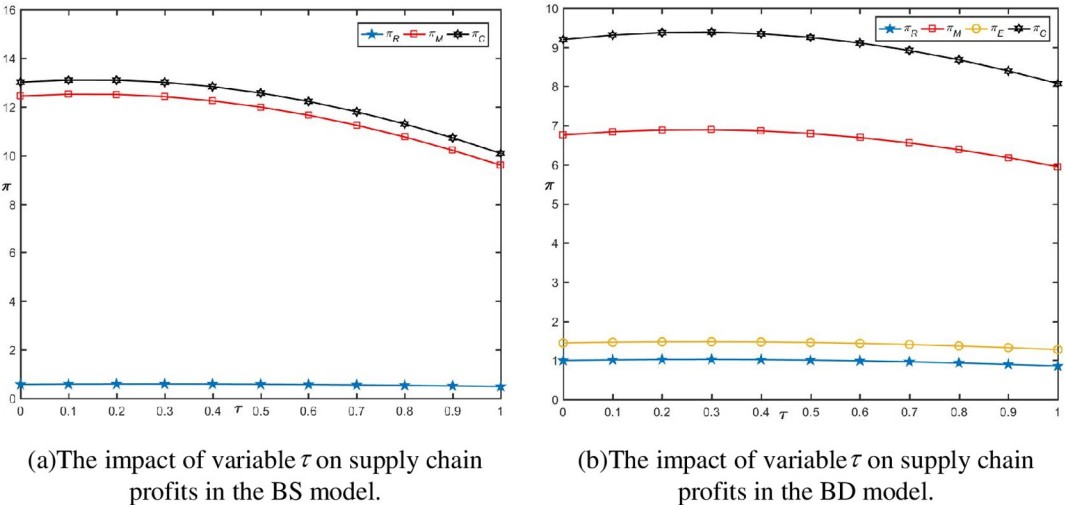

(a)The impact of variable $\tau$ on supply chain profits in the BS model.

(b)The impact of variable $\tau$ on supply chain profits in the BD model.

**Fig 3. The impact of variable $\tau$ on supply chain profits when adopting blockchain.**

From Fig 5, it is evident that in both modes, as consumers' preference for traditional channels increases, the retail and wholesale prices of traditional channels increase, while the retail and wholesale prices of online channels decrease. When the unit cost of blockchain is low, and consumers do not excessively prefer traditional channels. As shown in Fig 5(A), the following relationship exists for online channel retail prices: $p_e^{BD*} > p_e^{ND*} > p_e^{BS*} > p_e^{NS*}$; According to Fig 5(B), the following relationship exists for traditional channel retail prices: $p_r^{BD*} > p_r^{ND*} > p_r^{BS*} > p_r^{NS*}$; According to Fig 5(C), the following relationship exists for online

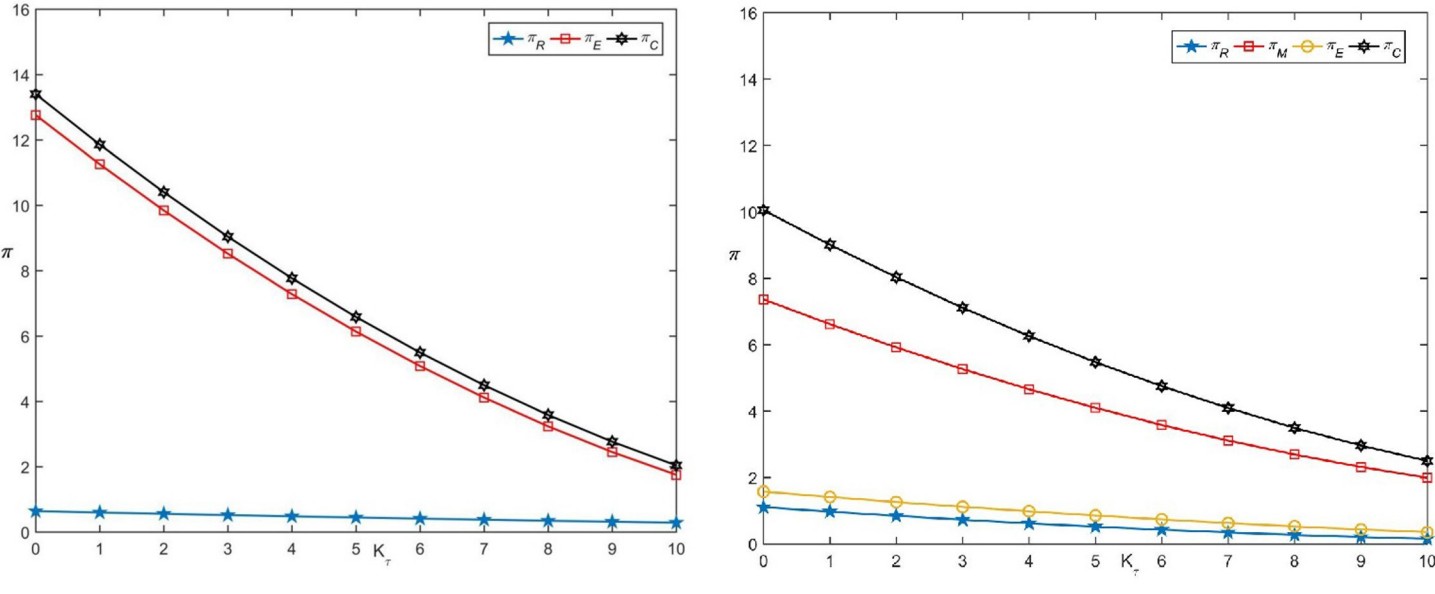

(a)The impact of variable $k_\tau$ on supply chain profits in the BS model.

(b)The impact of variable $k_\tau$ on supply chain profits in the BD model.

**Fig 4. The impact of variable $k_\tau$ on supply chain profits when adopting blockchain.**

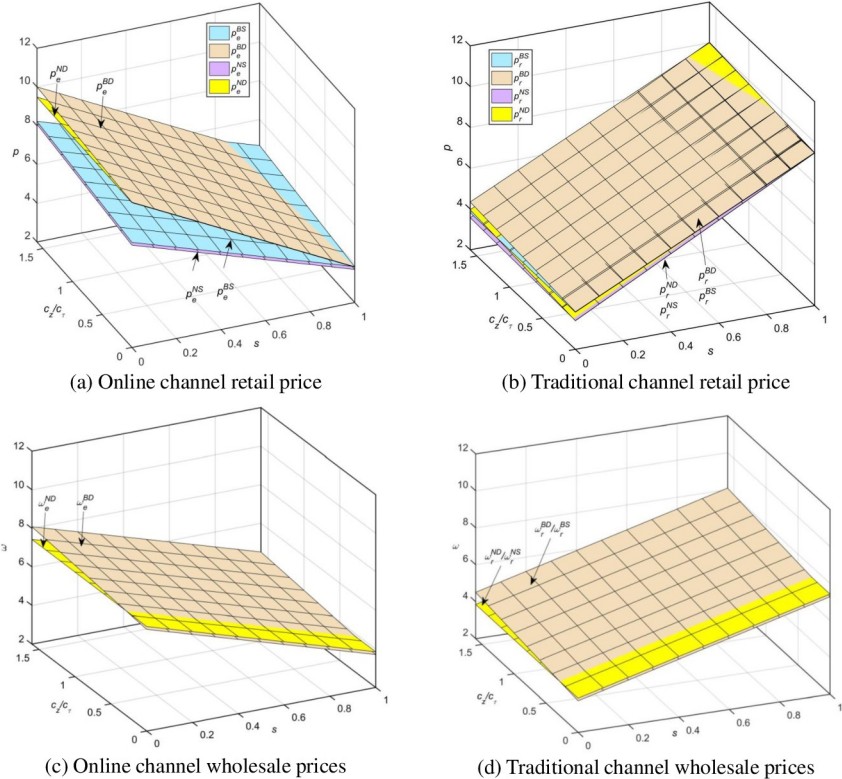

**Fig 5. Comparison of four models for supply chain pricing decisions.**

channel wholesale prices: $\omega_e^{BD*} > \omega_e^{ND*}$; According to Fig 5(D), the following relationship exists for traditional channel wholesale prices: $\omega_e^{BD*} = \omega_e^{BS*} > \omega_e^{ND*} = \omega_e^{NS*}$.

Fig 5 shows that regardless of the model, the pricing of traditional (online) channels increases with the degree of consumer preference for traditional (online) channels. When the unit cost of blockchain is low, and consumers do not excessively prefer traditional channels. Fig 5(A)–5(C) show that the channel pricing of fresh agricultural products manufacturers and traditional retailers under the online direct sales model is lower than the channel pricing in the online distribution model, and the pricing increases with the increase of direct sales costs or blockchain variable costs. This is because under the online direct sales model, fresh agricultural products manufacturers take advantage of their direct sales to reduce prices. This move will increase price competition with traditional retailers, so traditional retailers will also sell at lower retail prices to seize market share. Furthermore, under the same mode, various decision-making entities in the supply chain will, when adopting blockchain technology, transfer some costs to consumers by increasing the selling prices. Fig 5(D) shows that fresh agricultural products manufacturers give traditional retailers consistent wholesale prices under the online direct sales and online distribution models when (not) adopting blockchain technology. Generally speaking, when adopting blockchain technology, due to the increase in costs, fresh agricultural products manufacturers must transfer part of the blockchain technology costs to traditional retailers by increasing wholesale prices. But proposition 7 proves otherwise: when the unit cost of the blockchain is low, fresh agricultural products manufacturers have the ability to bear it alone, and even reduce the wholesale price to get more sales; When it is above a certain value, fresh agricultural products manufacturers will transfer part of the blockchain cost to retailers and consumers by raising wholesale prices and electronic retail prices.

When the variable cost of blockchain is below a certain threshold and consumers do not excessively prefer traditional channels, combining Fig 6(A) and 6(B), we can observe the following relationship for manufacturer profits: $\pi_M^{BS*} > \pi_M^{BD*} > \pi_M^{NS*} > \pi_M^{ND*}$; Traditional retailer profits exhibit the following relationship: $\pi_R^{BD*} > \pi_R^{BS*} > \pi_R^{ND*} > \pi_R^{NS*}$; According to Fig 6(C), the following relationship exists for total supply chain profits: $\pi_C^{BS*} > \pi_C^{BD*} > \pi_C^{ND*} > \pi_C^{NS*}$.

Fig 6 is further plotted using data to depict the trend of consumer channel preference and blockchain technology cost on manufacturer profits, retailer profits, total system profits and their value-added. The cost of blockchain technology has a greater impact on manufacturers' profits and system profits than on retailers. When the profits of all parties in the supply chain increase, the application and promotion of blockchain technology is the least difficult. If one party's profits are damaged and the other party's profits increase, it is necessary to establish a reasonable coordination mechanism, such as a cost-sharing mechanism, to ensure that both parties' profits increase and have motivation to promote application. If the profits of both parties are damaged, the government and other third-party authorities need to subsidize to ensure that the profits of both parties increase, and the difficulty of application and promotion will be moderately reduced. For example, the government supports the commodity traceability system built by AntChain in conjunction with Tmall Global and Cainiao through policy subsidies to trace the origin of Belgian diamonds, Australian imported milk, Wuchang rice and other commodities. Fig 6(B) shows that when blockchain technology is adopted, both manufacturers and traditional retailers have reduced their profits as the variable costs of blockchain increase; Regardless of the blockchain variable cost, manufacturers earn higher profits under the direct sales model than in the online distribution model, while traditional retailers have higher profits under the online distribution model.

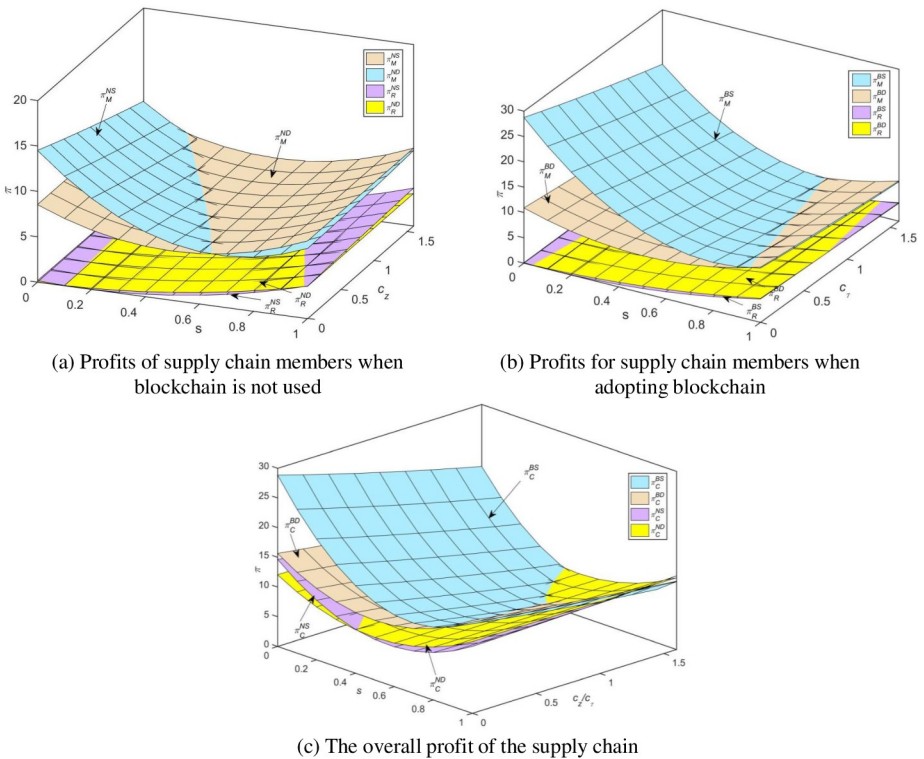

(a) Profits of supply chain members when blockchain is not used

(b) Profits for supply chain members when adopting blockchain

(c) The overall profit of the supply chain

**Fig 6. Comparison of profits of supply chain members under the four models.**

## 8 Conclusions

The characteristics of blockchain technology, such as decentralization, data immutability, and security transparency, can meet consumers' requirements for the safety and freshness of fresh agricultural products. This encourages retailers to not only seize the consumer market by opening up dual-channel models but also consider how to incorporate blockchain technology to achieve profit optimization. Additionally, the "smart contract" technology of blockchain can reduce transaction times, thereby decreasing double losses, while simultaneously ensuring the safety and transparency of fresh agricultural products, ultimately enhancing consumer trust. Given the dynamic coupling between blockchain unit variable costs and the trust gains it brings, this paper focuses on two common dual-channel supply chain systems: online direct sales and online distribution. It introduces parameters such as the level of blockchain technology investment, blockchain unit variable cost, and consumer channel preferences. Four scenarios are constructed: online direct sales and distribution channels without blockchain and online direct sales and distribution channels with blockchain. The paper analyzes the impact of blockchain and consumer channel preferences on dual-channel supply chain decisions. Based on this analysis, along with numerical simulations and sensitivity analysis of decision parameters, the following conclusions can be drawn.

1. Blockchain technology can effectively enhance the circulation efficiency of fresh agricultural products and the transparency of product information. On one hand, it alleviates the dual losses of fresh agricultural products during the circulation process, thereby reducing the comprehensive costs for manufacturers and facilitating the establishment of a situation with high-quality products at competitive prices. On the other hand, blockchain technology achieves traceability throughout the lifecycle of fresh agricultural products, reducing the "trust crisis" caused by information asymmetry. This, in turn, helps build consumer trust and expands the market demand for the fresh agricultural products supply chain.

2. Manufacturers opening online channels can seize market share and expand product demand. However, differences in consumer channel preferences and online sales models do not always guarantee profitability for manufacturers when opening online channels. Specifically, in the online direct sales model, when the manufacturer's online direct sales cost exceeds a certain threshold, the manufacturer's profit decreases with an increase in consumer preference for traditional channels and falls below the profit level in the distribution model. In the distribution model, with an increase in consumer preference for traditional channels, the manufacturer's profit exhibits a "U"-shaped variation.

3. When the variable cost of blockchain technology is relatively low, its adoption can significantly increase product pricing and the profits of various decision-making entities in the supply chain. This improvement becomes more pronounced as the degree of blockchain usage increases. Comparing the profits of various decision-making entities in the supply chain under two different dual-channel structural models reveals that when the variable cost of blockchain technology is relatively low, the manufacturer's profit and the overall system profit are both higher in the direct sales model than in the distribution model. However, the profit of traditional retailers in the direct sales model is lower than that in the distribution model. As the level of blockchain usage increases, dual-channel demand and overall supply chain profits exhibit an inverted "U"-shaped variation. This indicates that there is a peak point in supply chain profits, serving as the optimal investment point for blockchain technology.

The above conclusions provide insights into the application of blockchain in dual-channel supply chains for agricultural products. Firstly, blockchain technology allows consumers to access genuine product information, enabling them to identify product quality and obtain a better purchasing experience. This point is equally applicable to industries where accessing real information is difficult and discerning product authenticity is challenging. Secondly, this study examines the impact of the degree of blockchain usage and variable costs on the decision-making behavior of various participants in the fresh agricultural products supply chain. The results indicate that when the variable cost of blockchain is excessively high, the profits of decision-making entities in the fresh agricultural products supply chain will be lower than when not using blockchain, ultimately leading to a lack of motivation for its adoption. Therefore, manufacturers and retailers of fresh agricultural products can use blockchain benefit assessments and cost predictions to explore the conditions for implementing blockchain. Finally, adopting blockchain technology within a certain cost range can effectively reduce the dual losses of fresh agricultural products, increase consumer trust, alleviate channel conflicts, and thereby enhance the overall profitability of the supply chain. However, the high application cost of blockchain technology currently limits its adoption in the fresh supply chain. To promote the practical application of blockchain technology in relevant businesses, government authorities may consider providing corresponding policy incentives and welfare conditions. For example, subsidies could be offered to alleviate the investment costs for businesses.

This paper has analyzed, from the perspective of fresh agricultural product manufacturers as the dominant players, the impact of blockchain unit variable costs, direct selling costs, and consumer channel preferences on the pricing and channel selection in the dual-channel supply chain for online direct sales and online distribution of agricultural products. In the future, further discussions could explore the combined impact of blockchain technology and government subsidies, while also considering the influence on sustainable development when assessing social welfare.

## Supporting information

**S1 Appendix.**
(DOCX)

## Acknowledgments

We would like to thank the anonymous reviewers for their constructive comments.

## Author Contributions

**Conceptualization:** Di Wang, Xiaoyue Tian.

**Funding acquisition:** Di Wang.

**Methodology:** Di Wang.

**Validation:** Xiaoyue Tian, Mengchao Guo.

**Writing – original draft:** Di Wang, Xiaoyue Tian.

**Writing – review & editing:** Di Wang, Xiaoyue Tian, Mengchao Guo.

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
