## [Decision Letter · Decision Letter 0]

25 Jul 2023

PONE-D-23-21650Pricing Decision and Channel Selection of Fresh Agricultural Products Dual-channel Sup-ply Chain Based on BlockchainPLOS ONE

Dear Dr. Wang,

Thank you for submitting your manuscript to PLOS ONE. After careful consideration, we feel that it has merit but does not fully meet PLOS ONE’s publication criteria as it currently stands. Therefore, we invite you to submit a revised version of the manuscript that addresses the points raised during the review process.

We look forward to receiving your revised manuscript.

Kind regards,

Amir M. Fathollahi-Fard

Academic Editor

PLOS ONE

“Funding: Henan Province Philosophy and Social Science Planning Project（http://www.hnpopss.gov.cn/）Zhifang Li & Di Wang(2022CZH016); Henan Polytechnic University Basic Research Business Fund Special Project (http://www.hpu.edu.cn) Di Wang (SKJYB2023-13); Henan Polytechnic University Young Backbone Teacher Funding Scheme (http://www.hpu.edu.cn) Di Wang (2022XQG-14); The funders had no role in study design, data collection and analysis, decision to publish, or preparation of the manuscript.”

Additional Editor Comments:

Thank you for submitting your manuscript to PLOS ONE journal. Your paper underwent a thorough review by two expert reviewers, who provided valuable feedback. They have suggested major revisions, and I concur with most of their comments. As the academic editor, I would like to add some specific points for revision to enhance the quality and suitability of your paper for our journal.

Firstly, I would like to highlight that the current format of your manuscript does not adhere to our journal's guidelines for authors. Before proceeding with the revision, I kindly request that you review the guidelines and make necessary adjustments to ensure compliance.

Next, it is essential to revise the introduction section to more effectively elucidate the primary motivations, needs, and benefits of your research. This will provide readers with a clearer understanding of the significance of your paper.

The literature review in your manuscript overlooks several relevant articles in the field, which I recommend incorporating:

Asghari, M., Afshari, H., Mirzapour Al-e-hashem, S. M. J., Fathollahi-Fard, A. M., & Dulebenets, M. A. (2022). Pricing and advertising decisions in a direct-sales closed-loop supply chain. Computers & Industrial Engineering, 171, 108439.

Edalatpour, M. A., Mirzapour Al-e-Hashem, S. M. J., & Fathollahi-Fard, A. M. (2023). Combination of pricing and inventory policies for deteriorating products with sustainability considerations. Environment, Development and Sustainability, 1-41.

Fathollahi-Fard, A. M., Dulebenets, M. A., Hajiaghaei–Keshteli, M., Tavakkoli-Moghaddam, R., Safaeian, M., & Mirzahosseinian, H. (2021). Two hybrid meta-heuristic algorithms for a dual-channel closed-loop supply chain network design problem in the tire industry under uncertainty. Advanced engineering informatics, 50, 101418.

To clarify and justify the novelty of your work in comparison with these published works, I recommend providing a comparative table.

In Section 3, before subsection 3.1, please provide justifications and clarifications regarding the objectives of this section and the rationale for its division into different subsections. Additionally, establish links between these subsections. The same applies to Section 4 and Section 5.

Section 4 holds significant importance in your paper. However, it requires better presentation. Many formulations lack sufficient explanation, which may hinder readers' understanding. I suggest providing more detailed explanations to improve clarity.

Furthermore, it is essential for the authors to include a table comparing the models and incorporate charts to analyze the behavior of their models in a comparative study.

Finally, in the conclusion section, it is crucial to discuss the limitations of your research and propose potential areas for future research.

Please take these comments into consideration while revising your manuscript. Once you have completed the revisions, resubmit your paper, and it will undergo further evaluation.

Thank you for your attention to these matters, and I look forward to receiving the revised version of your manuscript.

Reviewers' comments:

Reviewer's Responses to Questions

**Comments to the Author**

1. Is the manuscript technically sound, and do the data support the conclusions?

Reviewer #1: Yes

Reviewer #2: Yes

2. Has the statistical analysis been performed appropriately and rigorously? 

Reviewer #1: Yes

Reviewer #2: Yes

3. Have the authors made all data underlying the findings in their manuscript fully available?

Reviewer #1: Yes

Reviewer #2: Yes

4. Is the manuscript presented in an intelligible fashion and written in standard English?

Reviewer #1: Yes

Reviewer #2: Yes

5. Review Comments to the Author

Reviewer #1: Reviewer Comments:

1. The paper addresses an important topic of pricing decision and channel selection in a dual-channel supply chain for fresh agricultural products based on blockchain technology. However, there are several areas that need improvement.

2. The motivation for the study is not clearly articulated. It would be helpful to provide a more compelling rationale for why this research is necessary and how it contributes to the existing literature. Additionally, the lack of innovation in the paper is evident as mainly focuses on applying blockchain technology to the existing supply chain without introducing any novel concepts or approaches.

3. The discussion section is quite brief and lacks depth. It would be beneficial to expand on the findings and provide a more comprehensive analysis of the results. Furthermore, the paper would benefit from a critical evaluation of the limitations and implications of the study.

4. The language and writing style of the paper need improvement. There are grammatical errors and awkward sentence structures throughout the manuscript. I recommend having the paper proofread by a native English speaker to enhance its readability and clarity.

Questions:

1. How does the proposed dual-channel structure differ from traditional supply chain models? What are the advantages and disadvantages of each model?

2. Can you elaborate on the methodology used to construct the four dual-channel supply chain decision models? How were the optimal strategies for pricing and channel selection determined?

3. The paper mentions that the introduction of blockchain technology can improve consumer trust and shorten circulation time. Can you provide more details on how blockchain achieves these outcomes in the context of the fresh agricultural products supply chain?

4. The paper states that the selling price rises with direct sales costs or blockchain change costs in both dual-channel structure models. Can you explain the underlying reasons for this relationship? How do these costs impact the profitability of the supply chain members?

5. The paper mentions that the ability of each member and system to bear the variable cost of blockchain technology varies. Could you elaborate on the factors that influence this ability and how it affects pricing and profitability?

6. Here are several recommended references to discuss in the literature and enhance its richness.

Bamakan, S. M. H., Malekinejad, P., & Ziaeian, M. (2022). Towards blockchain-based hospital waste management systems; applications and future trends. Journal of Cleaner Production, 131440.

Bamakan, S. M. H., Moghaddam, S. G., & Manshadi, S. D. (2021). Blockchain-enabled pharmaceutical cold chain: applications, key challenges, and future trends. Journal of Cleaner Production, 127021.

Bamakan, S. M. H., Faregh, N., & ZareRavasan, A. (2021). Di-ANFIS: an integrated blockchain–IoT–big data-enabled framework for evaluating service supply chain performance. Journal of Computational Design and Engineering, 8(2), 676-690.

Bamakan, S. M. H., Nezhadsistani, N., Bodaghi, O., & Qu, Q. (2022). Patents and intellectual property assets as non-fungible tokens; key technologies and challenges. Scientific Reports, 12(1), 1-13. Nature Publisher

Far, S. B., & Bamakan, S. M. H. (2022). Blockchain-based reporting protocols as a collective monitoring mechanism in DAOs. Data Science and Management, 5(1), 11-12.

Reviewer #2: Paper PONE-D-23-21650 “Pricing Decision and Channel Selection of Fresh Agricultural Products Dual-channel Sup-ply Chain Based on Blockchain?”

Comments

This study focuses on pricing decisions and channel selection of fresh agricultural products dual-channel sup-ply chain based on blockchain. I think the paper fits well the scope of the journal and addresses an important subject. However, a number of revisions are required before the paper can be considered for publication. There are some weak points that have to be strengthened. Below please find more specific comments:

*Abstract: The abstract should include a few preliminary sentences to highlight the importance of the topic. The authors start with the main concentration of the study immediately.

*Keywords: The keywords seem to be adequate. No comments.

*The introduction section could benefit from more statistical information to better highlight the importance of the main subject at hand.

*The literature review: please double check for the most recent and relevant studies published over the last 2-3 years. I see some recent and relevant studies on closed-loop supply chains and other important supply chain issues in general are missing, including but not limited to the following:

Supply chain disruption during the COVID-19 pandemic: Recognizing potential disruption management strategies. International Journal of Disaster Risk Reduction 2022, 75, p.102983.

Pricing and advertising decisions in a direct-sales closed-loop supply chain. Computers & Industrial Engineering 2022, 171, p.108439.

Two hybrid meta-heuristic algorithms for a dual-channel closed-loop supply chain network design problem in the tire industry under uncertainty. Advanced Engineering Informatics 2021, 50, p.101418.

Closed loop supply chains 4.0: From risks to benefits through advanced technologies. A literature review and research agenda. International Journal of Production Economics 2022, p.108582.

The relatedness of open-and closed-loop supply chains in the context of the circular economy; framing a continuum. Cleaner Logistics and Supply Chain 2022, p.100048.

It is essential that the literature review is up-to-date, so the relevant studies should be acknowledged.

*The model assumptions are presented well. I see that the adopted assumptions are supported by the relevant references. This will help justifying the adoption of these assumptions.

*The presentation of the proposed solution methodology seems to be adequate. I suggest adding more supporting references for the key mathematical relationships used.

*Please provide more details regarding the input data used throughout this study. More supporting references would be helpful to justify the data selection.

*The manuscript contains quite a lot of figures and tables. Please double check and try to provide a more detailed description of these figures and tables where appropriate to make sure that the future readers will have a reasonable understanding of what these figures and tables represent.

*The section devoted to numerical experiments should be expanded. It appears to be very short. The authors should include more analyses and discussions.

*Conclusions: The authors could expand more on the future research needs.

6. PLOS authors have the option to publish the peer review history of their article (what does this mean?). If published, this will include your full peer review and any attached files.

Reviewer #1: No

Reviewer #2: No

---

## [Author Response · Author response to Decision Letter 0]

6 Sep 2023

Response to Academic editor 

Dear academic editor,

Thank you very much for your kind review and suggestions. According to your suggestions, we have made careful modification of our manuscript. 

Thank you again for your positive comments and valuable suggestions to improve the quality of our manuscript.

With kind regards.

Di Wang

Point 1: Thank you for stating the following financial disclosure:

 “Funding: Henan Province Philosophy and Social Science Planning Project（http://www.hnpopss.gov.cn/）Zhifang Li & Di Wang(2022CZH016); Henan Polytechnic University Basic Research Business Fund Special Project (http://www.hpu.edu.cn) Di Wang (SKJYB2023-13); Henan Polytechnic University Young Backbone Teacher Funding Scheme (http://www.hpu.edu.cn) Di Wang (2022XQG-14); The funders had no role in study design, data collection and analysis, decision to publish, or preparation of the manuscript.”

Response 1: Thank you for your comment. We have added funding information to the latest cover letter. The detailed information is as follows:

(1) Funding Institution: Henan Provincial Department of Philosophy and Social Sciences Planning; Fund Project: Henan Provincial Department of Philosophy and Social Sciences Planning Project (2022CZH016) (http://www.hnpopss.gov.cn/); Project Leader: Di Wang.

(2) Funding Institution: Henan Polytechnic University; Fund Project: Henan Polytechnic University Basic Research Business Special Project (SKJYB2023-13) (http://www.hpu.edu.cn); Project Leader: Di Wang.

(3) Funding Institution: Henan Polytechnic University; Fund Project: Henan Polytechnic University Young Backbone Teacher Support Program (2022XQG-14) (http://www.hpu.edu.cn); Project Leader: Di Wang

The funder, Wang Di, as the first author and corresponding author of this article, took on data analysis, preparation of the manuscript, and publication decision-making responsibilities in this study.

Point 2: Please include captions for your Supporting Information files at the end of your manuscript, and update any in-text citations to match accordingly. 

Response 2: Thank you for your comment. We have revised captions for the Supporting Information files at the end of the manuscript, and updated all the in-text citations to match accordingly the manuscript as required in accordance with the journal’s guidelines.

Additional Editor Comments:

Point 1: Firstly, I would like to highlight that the current format of your manuscript does not adhere to our journal's guidelines for authors. Before proceeding with the revision, I kindly request that you review the guidelines and make necessary adjustments to ensure compliance.

Response 1: Thank you for your comment. We have revised the previous manuscript as required in accordance with the journal’s guidelines.

Point 2: Next, it is essential to revise the introduction section to more effectively elucidate the primary motivations, needs, and benefits of your research. This will provide readers with a clearer understanding of the significance of your paper.

Response 2: Thank you for your positive comments and valuable suggestions to improve the quality of our manuscript. In view of the main motivation, needs and significance of this study, we have revised the introduction section of this paper.

Point 3: The literature review in your manuscript overlooks several relevant articles in the field, which I recommend incorporating:

Asghari, M., Afshari, H., Mirzapour Al-e-hashem, S. M. J., Fathollahi-Fard, A. M., & Dulebenets, M. A. (2022). Pricing and advertising decisions in a direct-sales closed-loop supply chain. Computers & Industrial Engineering, 171, 108439.

Edalatpour, M. A., Mirzapour Al-e-Hashem, S. M. J., & Fathollahi-Fard, A. M. (2023). Combination of pricing and inventory policies for deteriorating products with sustainability considerations. Environment, Development and Sustainability, 1-41.

Fathollahi-Fard, A. M., Dulebenets, M. A., Hajiaghaei–Keshteli, M., Tavakkoli-Moghaddam, R., Safaeian, M., & Mirzahosseinian, H. (2021). Two hybrid meta-heuristic algorithms for a dual-channel closed-loop supply chain network design problem in the tire industry under uncertainty. Advanced engineering informatics, 50, 101418.

Response 3: Thank you for your comment and valuable suggestions. As suggested by the reviewer, we have added the relevant references to the manuscript. E.g.: Reference [16], [17] and [23].

Point 4: To clarify and justify the novelty of your work in comparison with these published works, I recommend providing a comparative table.

Response 4: Thank you for your comment and valuable suggestions. Based on your recommendations, we have compared this paper with the cited published articles, and the results are presented in Table 1 of the revised version.

Point 5: In Section 3, before subsection 3.1, please provide justifications and clarifications regarding the objectives of this section and the rationale for its division into different subsections. Additionally, establish links between these subsections. The same applies to Section 4 and Section 5.

Response 5: Thank you for your comment. We have added transitional paragraphs between the various sections of the article, elucidating the research objectives, content, and focal points of each section, serving as bridges between them.

Point 6: Section 4 holds significant importance in your paper. However, it requires better presentation. Many formulations lack sufficient explanation, which may hinder readers' understanding. I suggest providing more detailed explanations to improve clarity.

Response 6: Thanks for your comments and valuable suggestions. According to your suggestions, we have supplemented and modified the Section 4 of this article. For the formula solving part, we add a detailed solution process. We also give a more detailed explanation of the explanatory part of the proposition. Based on your feedback, we have made enhancements and revisions to Section 4 of this article. In the formula-solving segment, we have included a comprehensive step-by-step solution process. Additionally, we have provided a more in-depth explanation of the propositions.

Point 7: Furthermore, it is essential for the authors to include a table comparing the models and incorporate charts to analyze the behavior of their models in a comparative study.

Response 7: Thank you for your comment. In accordance with your recommendations, we have conducted a model comparison in Tables 3 and 4 within this article, presenting the optimal decisions for each model. Furthermore, we have incorporated Figures 2-4 into the numerical analysis section of the paper to investigate the effects of various parameters on the profits of supply chain participants across different models. Specifically, Figure 2-4 respectively delve into blockchain unit variable cost, the level of blockchain technology investment, and consumer channel preference on the profits of supply chain members.

Point 8: Finally, in the conclusion section, it is crucial to discuss the limitations of your research and propose potential areas for future research.

Response 8: Thank you for your comment. We have revised the conclusion section of this paper to discuss the limitations of this research and propose potential areas for future research.

Response to Reviewer 1 Comments

Dear reviewer,

Thank you very much for your kind review and suggestions. According to your suggestions, we have made careful modification of our manuscript. 

Thank you again for your positive comments and valuable suggestions to improve the quality of our manuscript.

With kind regards.

Di Wang

Point 1: The motivation for the study is not clearly articulated. It would be helpful to provide a more compelling rationale for why this research is necessary and how it contributes to the existing literature. Additionally, the lack of innovation in the paper is evident as mainly focuses on applying blockchain technology to the existing supply chain without introducing any novel concepts or approaches.

Response 1: Thank you for your comments on our article. At your suggestion, we have made supplementary revisions to the abstract, introduction, and conclusion sections to further elucidate the novelty and research significance of this article. 

The necessity and significance of this paper are primarily reflected in the following aspects: Most of the literature on the dual-channel supply chain of fresh agricultural products focuses on the impact of dual losses on supply chain decisions, while there is limited research on the pricing and channel selection in the context of fresh agricultural product supply chains based on blockchain technology. There is even less literature exploring the influence of blockchain technology on pricing and channel selection in dual-channel agricultural product supply chains. In the literature on the impact of blockchain technology on supply chain decisions, the discussion is limited to its effects on traditional single-channel supply chain decisions. Therefore, this paper selects two common dual-channel supply chain structures, online direct sales and online distribution, and combines specific parameters such as manufacturer's direct selling costs and consumer channel preferences. It further introduces blockchain-specific parameters such as the level of blockchain technology investment and blockchain variable costs. Four dual-channel agricultural product supply chain models are constructed, including two without blockchain technology and two with blockchain technology. These models are developed to analyze the optimal pricing and channel selection strategies under each mode using manufacturer-led Stackelberg game analysis. Finally, the rationality of the theoretical model is validated through numerical simulations. The aim is to provide theoretical guidance for the practical application of blockchain technology in the dual-channel supply chain of fresh agricultural products.

Point 2: The discussion section is quite brief and lacks depth. It would be beneficial to expand on the findings and provide a more comprehensive analysis of the results. Furthermore, the paper would benefit from a critical evaluation of the limitations and implications of the study.

Response 2: Thank you for your comments on our article. Based on your suggestions, in Section 6, we have provided a more comprehensive supplementation and exploration of the proposed propositions, and in Section 7, we have conducted a more detailed numerical simulation to evaluate and validate the impact of parameters such as the level of blockchain technology investment, blockchain variable costs, consumer channel preferences, etc., on supply chain decision-making. Simultaneously, we have expanded Section 8 of the article to summarize additional key findings and present relevant decision recommendations for different stakeholders. Furthermore, we have provided a forward-looking perspective on the limitations of this study.

Point 3: The language and writing style of the paper need improvement. There are grammatical errors and awkward sentence structures throughout the manuscript. I recommend having the paper proofread by a native English speaker to enhance its readability and clarity.

Response 3: Thank you for your comment and valuable suggestions. Based on your suggestions, we have proofread the entire article, corrected grammar errors, and adjusted sentence structures.

We have also responded to your questions below.

Question 1: How does the proposed dual-channel structure differ from traditional supply chain models? What are the advantages and disadvantages of each model?

Response 1: Thank you for your question. In this study, we introduce blockchain technology into a dual-channel agricultural supply chain dominated by manufacturers, considering parameters such as the level of blockchain technology investment, blockchain variable costs, and consumer channel preferences. We establish a Stackelberg game model to explore the impact of adopting blockchain and not adopting blockchain on pricing and channel decisions within the two dual-channel structures. We also conduct a comparative analysis between online direct sales and online distribution models. 

For the NS, ND, BS and BD models proposed in this paper, their advantages and disadvantages can be summarized as follows:

NS model: Under a certain online direct selling cost, manufacturers can through online channels to expand demand at a lower price, increase profits. However, this intensifies the channel competition with traditional retailers, which will lead to the loss of profits of traditional retailers.

ND model: Manufacturers using online distribution, relatively online direct sales model can reduce the competition between the two channels. However, this is not an optimal choice for the manufacturer himself.

BS model: The adoption of blockchain can improve and transform the existing enterprise network platform, thus saving the online direct selling cost of manufacturers. But in its place are the costs of blockchain technology. In addition, when the cost of blockchain technology is within a certain threshold, the profits of supply chain members and the overall can be effectively improved.

BD model: The adoption of blockchain technology will lead to an increase in the cost of manufacturers, but when the cost is within a certain threshold, compared with the ND model, the profits of supply chain members and the overall can be effectively improved under this model.

Question 2: Can you elaborate on the methodology used to construct the four dual-channel supply chain decision models? How were the optimal strategies for pricing and channel selection determined?

Response 2: Thank you for your question. In this article, we mainly use the method of game theory, guided by Stackelberg's game theory. Of these four models, a two-stage game is constructed, involving manufacturers and traditional and online retailers.

In the block before and after the chain technology builds the use of online direct marketing NS and BS model, the decision as follows: the first stage, the manufacturers in order to maximize their own interests as the goal, decision-making and traditional channels of the retail price of the online channel wholesale price. In the second stage, the traditional retailer decides the retail price in the traditional channel.

In the block before and after the chain technology builds the use of online distribution ND and BD model, the decision as follows: the first stage, the manufacturers in order to maximize their own interests as the goal, decides the wholesale price of online channel and traditional channel. In the second stage, the traditional and Internet retailers simultaneously decide the retail prices in their respective channels.

After the game model is constructed, the optimal pricing and profit under each model are obtained by backward induction. The best strategy for pricing and channel selection is then determined by parametric analysis and comparison between models. For example, by comparing NS, ND, BS and BD models respectively in 4.3 and 5.3 of this paper, the cost critical values of pricing and channel selection of manufacturers and retailers under the two models can be obtained. It is then possible to determine which channel model manufacturers should choose when the relevant costs are within the threshold and how manufacturers and retailers should price.

Question 3: The paper mentions that the introduction of blockchain technology can improve consumer trust and shorten circulation time. Can you provide more details on how blockchain achieves these outcomes in the context of the fresh agricultural products supply chain?

Response 3: Thank you for your question. Our answer to your question is as follows: The characteristics of blockchain technology, such as decentralization, data immutability, and security transparency, can meet consumers' requirements for the safety and freshness of fresh agricultural products. This encourages retailers to not only seize the consumer market by opening up dual-channel models but also consider how to incorporate blockchain technology to achieve profit optimization. Additionally, the "smart contract" technology of blockchain can reduce transaction times, thereby decreasing double losses, while simultaneously ensuring the safety and transparency of fresh agricultural products, ultimately enhancing consumer trust.

Question 4: The paper states that the selling price rises with direct sales costs or blockchain change costs in both dual-channel structure models. Can you explain the underlying reasons for this relationship? How do these costs impact the profitability of the supply chain members?

Response 4: Thank you for your comment. The underlying reasons for this relationship is that as the cost of direct sales or blockchain increases, manufacturers and retailers need to transfer part of the cost to ensure their own interests, so they need to transfer the cost to consumers by increasing the selling price.

When blockchain technology is not adopted, when manufacturers open up online direct sales channels for online sales, they have certain direct sales advantages in the online direct sales model compared with the network distribution model, and can conduct online sales at a lower price. At this time, when the cost of direct sales is small, the manufacturer's network channel can form a situation of small profits and high sales, thereby increasing its profitability. At the same time, due to the lower cost of direct sales, manufacturers will not pass on costs to traditional retailers by raising wholesale prices, but because the low-price sales behavior of manufacturers' network channels will shift some consumers from offline to online, it will make traditional retailers less profitable.

Similarly, when the variable cost of blockchain is high, fresh agricultural products manufacturers will transfer part of the cost to downstream retailers and consumers by increasing wholesale prices and electronic sales prices, and retailers will also transfer part of the costs to consumers again by increasing the dual-channel selling price. In the end, the high premium leads to a decrease in the demand of the dual channels, and the positive effect of the increase in net benefit per unit is less than the negative effect caused by the decrease in demand, thus reducing the profit capacity of both parties.

Question 5: The paper mentions that the ability of each member and system to bear the variable cost of blockchain technology varies. Could you elaborate on the factors that influence this ability and how it affects pricing and profitability?

Response 5: Thank you for your comment. In this paper, we first assume that the blockchain technology variable cost is solely borne by the manufacturer. In the process of the game, the manufacturer transfers the variable cost of blockchain technology to the traditional retailer by increasing the wholesale price, and the traditional retailer can transfer the cost to the consumer by increasing the retail price. Therefore, when the variable cost of blockchain technology is within a certain range, the profit of supply chain members can be effectively improved. However, different dual-channel structures and consumer channel preferences will change the ability of each member and system to bear the variable costs of blockchain technology.

Different dual-channel structures affect the ability of supply chain members to bear blockchain variable costs. For example, since the manufacturer has certain price and channel advantages under the online direct selling mode, the manufacturer has a stronger ability to bear the variable cost of blockchain technology under the online direct selling mode.

Consumer channel preferences also affect the ability of supply chain members to bear the variable costs of blockchain technology. For example, when consumers prefer traditional channels, because traditional retailers have greater channel advantages, the increase in demand and retail price will increase profits. Therefore, when manufacturers adopt blockchain technology, traditional retailers are better able to share.

Response to Reviewer 2 Comments

Dear reviewer,

Thank you very much for your kind review and suggestions. According to your suggestions, we have made careful modification of our manuscript. 

Thank you again for your positive comments and valuable suggestions to improve the quality of our manuscript.

With kind regards.

Di Wang

Point 1: The abstract should include a few preliminary sentences to highlight the importance of the topic. The authors start with the main concentration of the study immediately.

Response 1: Thank you for your positive comments and valuable suggestions to improve the quality of our manuscript. We have revised the abstract part of this paper based on the motivation and innovation points of this study.

Point 2: The introduction section could benefit from more statistical information to better highlight the importance of the main subject at hand.

Response 2: Thank you for your positive comments. We have added some statistics to the introduction of the article, the specific details of which are in the revised manuscript.

Point 3: The literature review: please double check for the most recent and relevant studies published over the last 2-3 years.

Response 3: Thank you for the comment. We have added the latest references to the revised manuscript.

Point 4: The model assumptions are presented well. I see that the adopted assumptions are supported by the relevant references. This will help justifying the adoption of these assumptions.

Response 4: Thank you for your nice comment.

Point 5: The presentation of the proposed solution methodology seems to be adequate. I suggest adding more supporting references for the key mathematical relationships used.

Response 5: Thank you for your comments and suggestions. In the revised version we have added more supporting references for the key mathematical relationships used.

Point 6: Please provide more details regarding the input data used throughout this study. More supporting references would be helpful to justify the data selection.

Response 6: Thank you for your comments and suggestions. We have revised the numerical analysis section of this article and added relevant references.

Point 7: The manuscript contains quite a lot of figures and tables. Please double check and try to provide a more detailed description of these figures and tables where appropriate to make sure that the future readers will have a reasonable understanding of what these figures and tables represent.

Response 7: Thank you for the comment. We have examined and revised the figures and tables that appear in the text, as can be found in the revised manuscript.

Point 8: The section devoted to numerical experiments should be expanded. It appears to be very short. The authors should include more analyses and discussions.

Response 8: Thank you for your comments and suggestions. We have added Figures 2-4 in the numerical analysis section of this paper to study the impact of relevant parameters on the profits of supply chain members under different models. Among them, figure 2-4 respectively when the other parameters must be explored, consumer preference for traditional channels, the level of blockchain technology investment, blockchain technology cost sensitivity influence on profit of supply chain members. expanded and discussed the numerical experiments section of this article.

---

## [Decision Letter · Decision Letter 1]

15 Dec 2023

PONE-D-23-21650R1Pricing Decision and Channel Selection of Fresh Agricultural Products Dual-channel Supply Chain Based on BlockchainPLOS ONE

Dear Dr. Wang,

Thank you for submitting your manuscript to PLOS ONE. After careful consideration, we feel that it has merit but does not fully meet PLOS ONE’s publication criteria as it currently stands. Therefore, we invite you to submit a revised version of the manuscript that addresses the points raised during the review process.

We look forward to receiving your revised manuscript.

Kind regards,

Vanessa Carels

Staff Editor

PLOS ONE

**Additional Editor Comments:**

This revised submission has been assessed by a number of reviewers, and their comments are available below. The reviewers have raised a number of concerns that need attention.  Could you please revise the manuscript to carefully address the concerns raised?

Reviewers' comments:

Reviewer's Responses to Questions

**Comments to the Author**

1. If the authors have adequately addressed your comments raised in a previous round of review and you feel that this manuscript is now acceptable for publication, you may indicate that here to bypass the “Comments to the Author” section, enter your conflict of interest statement in the “Confidential to Editor” section, and submit your "Accept" recommendation.

Reviewer #1: All comments have been addressed

Reviewer #2: All comments have been addressed

Reviewer #3: (No Response)

Reviewer #4: (No Response)

Reviewer #5: (No Response)

Reviewer #6: (No Response)

2. Is the manuscript technically sound, and do the data support the conclusions?

Reviewer #1: Yes

Reviewer #2: Yes

Reviewer #3: (No Response)

Reviewer #4: Partly

Reviewer #5: Yes

Reviewer #6: (No Response)

3. Has the statistical analysis been performed appropriately and rigorously? 

Reviewer #1: Yes

Reviewer #2: Yes

Reviewer #3: (No Response)

Reviewer #4: N/A

Reviewer #5: Yes

Reviewer #6: (No Response)

4. Have the authors made all data underlying the findings in their manuscript fully available?

Reviewer #1: Yes

Reviewer #2: Yes

Reviewer #3: (No Response)

Reviewer #4: Yes

Reviewer #5: Yes

Reviewer #6: (No Response)

5. Is the manuscript presented in an intelligible fashion and written in standard English?

Reviewer #1: Yes

Reviewer #2: Yes

Reviewer #3: (No Response)

Reviewer #4: Yes

Reviewer #5: Yes

Reviewer #6: (No Response)

6. Review Comments to the Author

Reviewer #1: After carefully reviewing the revised paper, it appears that the authors have made significant improvements to address the issues raised in the previous review. The revised paper now provides a more comprehensive analysis of the research problem and presents a clearer and more coherent argument. Additionally, the authors have provided detailed responses to all the questions raised by the reviewers, demonstrating a thorough understanding of the research area and a willingness to engage with feedback. Overall, the revisions have greatly strengthened the paper and it is now ready for publication.

Reviewer #2: The authors took seriously my previous comments and made the required revisions in the manuscript. The quality and presentation of the manuscript have been improved. Therefore, I recommend acceptance.

Reviewer #3: Since the comments includes fomula, all the comments are also upload as an attachment.

The manuscript shows several flaws.

There are several empty cells in table 1 and it doesn't look nice. Is it appropriate to fill in the left slash? Please adjust the width of each column of the table 1. For example, the first column should be narrower, while the second column may be wider.

The font of the word “section” on line 191 is inconsistent with others.

Fig 1 shows some flaws. The most fatal point is that the third and fourth subfigures fail to show any difference from the first and second subfigures. I suggest that the author include text boxes and arrows in the latter two subfigures to show the adoption of blockchain. Then, the four subfigures (a), (b), (c), (d) are not in Times New Roman. Last, w_r^BD in the fourth subfigure is bolded, while others are not.

In line 253, the function includes exponential terms e^(ln2/T t). For better readability, I suggest replacing it with exp⁡(ln2/T t).

The first column of table 3 shows NS*, while the third column is ND model. Similar problem exists in table 4.

Fig 5 shows weak resolvability. Please revise it to make the color difference more noticeable. Even it is obvious of Fig 6, please fill 3 different colors. Also, the size should be revised.

Some references are wrong.

Journal names are not all capitalized of 4, 13, 17, 23.

All words of the title are capitalized of 6, 38, 43, 59, 60.

Journal name has extra space of 8.

There is no uppercase after the colon of 9, 37.

The journal name is wrong of 12, 26, 30, and computer should be computers.

& is used in some references to link authors (for example 14), but others are not (for example 15).

The journal's name is abbreviated of 22.

Colon of the journal name is missing in 42.

Reviewer #4: Point 1: The statement of the research problem in the introduction is not clear. Judging from the title, the research objective of this paper should be pricing strategy and channel selection, but the paper does not explain why it needs to conduct pricing strategy and channel selection, and the practical and theoretical significance of the research is not clear (Does the producer company in reality have a practical demand for price decision and channel responsibility based on blockchain technology’s dual-channel supply chain? Are there deficiencies and gaps in existing research in this area?). The innovative points and contributions of the research in the introduction are not clearly explained.

Point 2: The statement of the research problem in the introduction is not clear. Judging from the title, the research objective of this paper should be pricing strategy and channel selection, but the paper does not explain why it needs to conduct pricing strategy and channel selection, and the practical and theoretical significance of the research is not clear (Does the producer company in reality have a practical demand for price decision and channel responsibility based on blockchain technology’s dual-channel supply chain? Are there deficiencies and gaps in existing research in this area?). The innovative points and contributions of the research in the introduction are not clearly explained.

Point 3: The assumptions of the model in Chapter 3 are the same as the dual-channel model in reference [43]. It is recommended to clarify the differences between the existing study [43] and this research in the literature review section, which could highlight the innovation and contributions of this paper.

Point 4: What assumption from reference [27] is referenced in the numerical analysis section? What actual situation is it based on? Please provide a detailed explanation, as this will demonstrate the rationality of the numerical analysis.

Point 5: The conclusion of the article is almost consistent with the results of reference [43], offering no new findings, and fails to reflect the innovation and contribution of this study in the theoretical aspect.

Reviewer #5: (No Response)

Reviewer #6: Article “Pricing Decision and Channel Selection of Fresh Agricultural Products Dual-channel Supply Chain Based on Blockchain”（PONE-D-23-21650）has been basically modified according to expert opinions. But there are still some problems, mainly as follows:

1. The impact of information asymmetry in the supply chain on fresh produce is not clearly described in the introduction.

2. In the literature review section, it is recommended to add a table that compares the existing research with the research focus of this article to highlight the novelty of the article.

3. The model description is not clear, and the circulation time of the product should be t0 , when blockchain technology is not used. And please explain why the circulation time of the product changes after using blockchain technology.

4. Whether the parameter settings in the numerical study have a corresponding practical basis or theoretical support.

5. The conclusion of the article is less content, and it is recommended to enrich the relevant content.

7. PLOS authors have the option to publish the peer review history of their article (what does this mean?). If published, this will include your full peer review and any attached files.

Reviewer #1: No

Reviewer #2: No

Reviewer #3: No

Reviewer #4: No

Reviewer #5: No

Reviewer #6: No

---

## [Author Response · Author response to Decision Letter 1]

29 Dec 2023

Response to Reviewer #3 Comments 

Dear reviewer,

Thank you very much for your kind review and suggestions. According to your suggestions, we have made careful modification of our manuscript. 

Thank you again for your positive comments and valuable suggestions to improve the quality of our manuscript.

With kind regards.

Di Wang

 There are several empty cells in table 1 and it doesn't look nice. Is it appropriate to fill in the left slash? Please adjust the width of each column of the table 1. For example, the first column should be narrower, while the second column may be wider.

Response 1: Thank you for your comments. We have filled every empty cells in table 1 with a left slash and adjusted the width of each column to make the table 1 look more comfortable.

 The font of the word “section” on line 191 is inconsistent with others. 

Response 2: Thank you very much for pointing out this problem, we have changed the font of the word “section” on line 191 to Times New Roman used in this article.

 Fig 1 shows some flaws. The most fatal point is that the third and fourth subfigures fail to show any difference from the first and second subfigures. I suggest that the author include text boxes and arrows in the latter two subfigures to show the adoption of blockchain. Then, the four subfigures (a), (b), (c), (d) are not in Times New Roman. Last, w_r^BD in the fourth subfigure is bolded, while others are not.

Response 3: Thanks for your comments, we have fixed the defect in Fig 1. First, we added blockchain technology section at the top of the latter two subfigures in the form of text boxes and arrows to show the adoption of blockchain. Second, we change the four subfigures (a), (b), (c), and (d) to Times New Roman. Finally, the bold form of w_r^BD in the fourth subfigure is also removed to keep it consistent with others.

 In line 253, the function includes exponential terms e^(ln2/T t). For better readability, I suggest replacing it with exp⁡(ln2/T t).

Response 4: Thank you very much for your comments, and we have adopted your suggestion to replace the function includes exponential terms e^(ln2/T t) in line 253 with exp⁡(ln2/T t).

 The first column of table 3 shows NS*, while the third column is ND model. Similar problem exists in table 4.

Response 5: Thank you very much for your comments. We have modified the symbols in the first column of table 3 and table 4 accordingly.

 Fig 5 shows weak resolvability. Please revise it to make the color difference more noticeable. Even it is obvious of Fig 6, please fill 3 different colors. Also, the size should be revised.

Response 6: Thank you for your suggestions. We have modified Fig 5 and Fig 6 according to your suggestions in the revised manuscript.

 Some references are wrong.

 Journal names are not all capitalized of 4, 13, 17, 23. 

 All words of the title are capitalized of 6, 38, 43, 59, 60. 

 Journal name has extra space of 8. 

 There is no uppercase after the colon of 9, 37. 

 The journal name is wrong of 12, 26, 30, and computer should be computers.

 & is used in some references to link authors (for example 14), but others are not (for example 15).

 The journal's name is abbreviated of 22.

 Colon of the journal name is missing in 42.

Response 7: Thank you very much for your suggestions. We have revised the problems in the references you raised one by one.

 

Response to Reviewer #4 Comments 

Dear reviewer,

Thank you very much for your kind review and suggestions. According to your suggestions, we have made careful modification of our manuscript. 

Thank you again for your positive comments and valuable suggestions to improve the quality of our manuscript.

With kind regards.

Di Wang

Point 1: The statement of the research problem in the introduction is not clear. Judging from the title, the research objective of this paper should be pricing strategy and channel selection, but the paper does not explain why it needs to conduct pricing strategy and channel selection, and the practical and theoretical significance of the research is not clear (Does the producer company in reality have a practical demand for price decision and channel responsibility based on blockchain technology’s dual-channel supply chain? Are there deficiencies and gaps in existing research in this area?). The innovative points and contributions of the research in the introduction are not clearly explained.

Response 1: Thank you for your comments. According to your suggestions, we have made modifications to the introduction and literature review sections of this paper. In the introduction, we explain the reasons for conducting research on pricing strategies and channel selection, and have included company examples to clarify the significance, innovation, and contribution of this study. In the literature review section, we provide an overview of research on pricing decisions and channel selection in the dual-channel supply chain of fresh agricultural products, summarizing the differences between this paper and existing literature. The specific reasons, significance, and innovative aspects of this paper's study on the pricing strategies and channel selection of the dual-channel supply chain for fresh agricultural products under blockchain technology are as follows.

With the development of the online retail market, the integration of online and offline is becoming increasingly mature. Fresh agricultural product manufacturers, especially those in agricultural industrialization and professional cooperatives, to expand their presence beyond traditional retail channels by venturing into the online sphere, aiming to enhance the quality of agricultural products and capture a larger market share. High-quality agricultural product manufacturers such as Anchor, Dole, and Jiawo adopt the online direct sales model, while some smaller-scale agricultural product manufacturers choose the online distribution model. The sales model based on a dual-channel structure will reduce the sales costs for enterprises and increase their market share, thereby benefiting the business. However, the diversification of dual-channel structures, intense channel competition, easily trigger "free-riding" behavior and vicious price competition. In addition, consumer channel preferences can also have a significant impact on the management decisions of supply chain enterprises, such as influencing the price competition relationships among businesses. In this sense, it is crucial to study the pricing and channel selection issues of the dual-channel supply chain of fresh agricultural products composed of manufacturers and retailers from a systemic perspective.

Additionally, through the analysis of relevant literature, we have found that existing studies often approach blockchain technology and the fresh agricultural product supply chain from a singular perspective, with limited literature quantifying the impact of blockchain technology. Therefore, this paper focuses on a second-tier fresh agricultural product supply chain composed of individual suppliers and retailers. It considers factors such as the circulation efficiency of fresh agricultural products, blockchain unit variable costs, the level of investment in blockchain technology, and consumer channel preferences. The paper employs methods such as model construction and numerical analysis to study the pricing and channel selection issues in a dual-channel supply chain for fresh agricultural products based on blockchain technology. The research conclusions can provide management and decision-making recommendations for different supply chain participants and offer insights and methods for addressing the aforementioned issues.

In the literature review section, we conducted a comparative analysis of the achievements in pricing decisions and channel selection in the dual-channel supply chain for fresh agricultural products. However, there are still gaps in the existing literature in the following aspects. Firstly, existing research has focused on pricing and coordination issues in single-channel or single dual-channel mode supply chains for fresh agricultural products, and there is a need for additional research on pricing and channel selection in dual-channel supply chains for fresh agricultural products under different dual-channel structural models. Secondly, most scholars have concentrated on case studies to explore the impact of blockchain on business operations and decision-making. There is a lack of models quantifying the economic benefits of blockchain technology in improving the circulation time of fresh agricultural products and increasing consumer trust under blockchain models. Thirdly, existing literature primarily studies the impact of blockchain technology on the supply chain from a singular perspective, with relatively few results considering the combined influence of consumer preferences, the level of blockchain utilization, and blockchain variable costs on simulation outcomes.

Point 2: The statement of the research problem in the introduction is not clear. Judging from the title, the research objective of this paper should be pricing strategy and channel selection, but the paper does not explain why it needs to conduct pricing strategy and channel selection, and the practical and theoretical significance of the research is not clear (Does the producer company in reality have a practical demand for price decision and channel responsibility based on blockchain technology’s dual-channel supply chain? Are there deficiencies and gaps in existing research in this area?). The innovative points and contributions of the research in the introduction are not clearly explained.

Response 2: Thank you for your comments. According to your suggestions, we have made modifications to the introduction and literature review sections of this paper. In the introduction, we explain the reasons for conducting research on pricing strategies and channel selection, and have included company examples to clarify the significance, innovation, and contribution of this study. In the literature review section, we provide an overview of research on pricing decisions and channel selection in the dual-channel supply chain of fresh agricultural products, summarizing the differences between this paper and existing literature. The specific reasons, significance, and innovative aspects of this paper's study on the pricing strategies and channel selection of the dual-channel supply chain for fresh agricultural products under blockchain technology are as follows.

With the development of the online retail market, the integration of online and offline is becoming increasingly mature. Fresh agricultural product manufacturers, especially those in agricultural industrialization and professional cooperatives, to expand their presence beyond traditional retail channels by venturing into the online sphere, aiming to enhance the quality of agricultural products and capture a larger market share. High-quality agricultural product manufacturers such as Anchor, Dole, and Jiawo adopt the online direct sales model, while some smaller-scale agricultural product manufacturers choose the online distribution model. The sales model based on a dual-channel structure will reduce the sales costs for enterprises and increase their market share, thereby benefiting the business. However, the diversification of dual-channel structures, intense channel competition, easily trigger "free-riding" behavior and vicious price competition. In addition, consumer channel preferences can also have a significant impact on the management decisions of supply chain enterprises, such as influencing the price competition relationships among businesses. In this sense, it is crucial to study the pricing and channel selection issues of the dual-channel supply chain of fresh agricultural products composed of manufacturers and retailers from a systemic perspective.

Additionally, through the analysis of relevant literature, we have found that existing studies often approach blockchain technology and the fresh agricultural product supply chain from a singular perspective, with limited literature quantifying the impact of blockchain technology. Therefore, this paper focuses on a second-tier fresh agricultural product supply chain composed of individual suppliers and retailers. It considers factors such as the circulation efficiency of fresh agricultural products, blockchain unit variable costs, the level of investment in blockchain technology, and consumer channel preferences. The paper employs methods such as model construction and numerical analysis to study the pricing and channel selection issues in a dual-channel supply chain for fresh agricultural products based on blockchain technology. The research conclusions can provide management and decision-making recommendations for different supply chain participants and offer insights and methods for addressing the aforementioned issues.

In the literature review section, we conducted a comparative analysis of the achievements in pricing decisions and channel selection in the dual-channel supply chain for fresh agricultural products. However, there are still gaps in the existing literature in the following aspects. Firstly, existing research has focused on pricing and coordination issues in single-channel or single dual-channel mode supply chains for fresh agricultural products, and there is a need for additional research on pricing and channel selection in dual-channel supply chains for fresh agricultural products under different dual-channel structural models. Secondly, most scholars have concentrated on case studies to explore the impact of blockchain on business operations and decision-making. There is a lack of models quantifying the economic benefits of blockchain technology in improving the circulation time of fresh agricultural products and increasing consumer trust under blockchain models. Thirdly, existing literature primarily studies the impact of blockchain technology on the supply chain from a singular perspective, with relatively few results considering the combined influence of consumer preferences, the level of blockchain utilization, and blockchain variable costs on simulation outcomes.

Point 3: The assumptions of the model in Chapter 3 are the same as the dual-channel model in reference [43]. It is recommended to clarify the differences between the existing study [43] and this research in the literature review section, which could highlight the innovation and contributions of this paper.

Response 3: Thank you for your suggestions, and we have clarified the differences between this article and reference [43] in the literature review section of the revised manuscript. This is specifically reflected in the following three aspects:

 (1) The research focus of this article differs from that of reference [43]. Our study primarily targets the supply chain of fresh agricultural products, introducing relevant parameters closely associated with the characteristics of fresh agricultural products, such as circulation time, freshness, and the ratio of effective output.

 (2) Unlike reference [43], this article comprehensively considers factors such as cross-price elasticity, consumer channel preferences, and online direct selling costs, all of which influence decision-making in dual-channel supply chains.

 (3) While reference [43] only considers the fixed costs associated with the introduction of blockchain technology, this article introduces the degree of blockchain usage. Furthermore, we incorporate blockchain technology's variable costs and consumer trust gains, emphasizing their significant impact on the adoption decision of blockchain technology.

Point 4: What assumption from reference [27] is referenced in the numerical analysis section? What actual situation is it based on? Please provide a detailed explanation, as this will demonstrate the rationality of the numerical analysis.

Response 4: Thank you for your comments. Special note: Due to revisions made to the article, reference [27] has been updated to reference [34]. This paper mainly cites the data of circulation time, production and transportation cost of fresh agricultural products in reference [34], and these data are based on the actual data and materials of a cherry manufacturer in Yantai, Shandong, China. We have made a detailed supplementary explanation of this part in the revised manuscript.

Point 5: The conclusion of the article is almost consistent with the results of reference [43], offering no new findings, and fails to reflect the innovation and contribution of this study in the theoretical aspect.

Response 5: Thank you for your comments. Based on the questions you raised, we have further summarized and supplemented the conclusion section of this paper. Additionally, we have presented some managerial insights derived from the research conclusions, enhancing the richness of the conclusion section.

 

Response to Reviewer #6 Comments 

Dear reviewer,

Thank you very much for your kind review and suggestions. According to your suggestions, we have made careful modification of our manuscript. 

Thank you again for your positive comments and valuable suggestions to improve the quality of our manuscript.

With kind regards.

Di Wang

1. The impact of information asymmetry in the supply chain on fresh produce is not clearly described in the introduction.

Response 1: Thanks for your comments. According to your suggestion, we have made a supplementary explanation on the impact of information asymmetry in the supply chain on fresh agricultural products in the introduction of this paper. With the circulation of products, product quality information will continue to decay, resulting in serious information asymmetry in the process of product quality supervision. Consumers in the inferior position of information can not trace the product quality, and the rights and interests of consumers can not be protected. In the traditional mode, the information of product quality traceability comes from the manufacturer or the third party enterprise. They may arbitrarily change the product information in order to pursue the maximization of benefits, which leads to the failure of consumers to effectively identify the product quality, and thus leads to the "trust crisis" of consumers on the disclosure of information by enterprises.

2. In the literature review section, it is recommended to add a table that compares the existing research with the research focus of this article to highlight the novelty of the article.

Response 2: Thank you for your suggestions. We have conducted a comparative analysis of relevant literature, and the results are presented in Table 1. Through this comparative analysis, we have identified several shortcomings in existing literature:

Firstly, previous research has mainly focused on pricing and coordination issues in the supply chain of fresh agricultural products under single-channel or single dual-channel models. There is a need for additional studies addressing the pricing and channel selection problems in the dual-channel supply chain of fresh agricultural products under different dual-channel structural models.

Secondly, most scholars have concentrated on using case studies to explore the impact of blockchain on enterprise operations and decision-making. However, there is a lack of models quantifying the economic benefits of blockchain technology in improving the circulation time of fresh agricultural products and increasing consumer trust under a blockchain paradigm.

Thirdly, existing literature primarily examines the impact of blockchain technology on the supply chain from a singular perspective. There is relatively less research that comprehensively considers the impact of consumer preferences, the degree of blockchain usage, and the variable costs of blockchain on simulation results.

In this paper, we focus on a dual-channel supply chain of fresh agricultural products consisting of one supplier and one retailer. We introduce parameters such as the circulation efficiency of fresh agricultural products, variable costs of blockchain units, the level of investment in blockchain technology, and consumer channel preferences. Using the Stackelberg game model, we conduct a comparative analysis of the pricing and channel selection strategies in the dual-channel supply chain of fresh agricultural products under two scenarios: without using blockchain technology and adopting blockchain technology.

3. The model description is not clear, and the circulation time of the product should be t0 , when blockchain technology is not used. And please explain why the circulation time of the product changes after using blockchain technology.

Response 3: Thanks for your comments and we have made corresponding modifications in Assumption 1. In this assumption, the circulation time of the products is assumed to be , when blockchain technology is not used. After the adoption of blockchain technology, mainly through the traceability system of blockchain, automatic identification of items and automatic collection of data can be successfully completed. Therefore, it can speed up the speed of logistics links, reduce invalid paths, and shorten the circulation time of products in upstream and downstream enterprises. Accordingly, the circulation time will change from the original to , , effectively improve the freshness of the product and effective output ratio.

4. Whether the parameter settings in the numerical study have a corresponding practical basis or theoretical support.

Response 4: Thank you for your comments. We have added a corresponding practical basis for parameter setting in numerical research in the revised manuscript.

5. The conclusion of the article is less content, and it is recommended to enrich the relevant content.↳

Response 5: Thanks for your comments, we have made corresponding contents to the conclusion of this article in the revised manuscript.

---

## [Decision Letter · Decision Letter 2]

8 Jan 2024

Pricing Decision and Channel Selection of Fresh Agricultural Products Dual-channel Supply Chain Based on Blockchain

PONE-D-23-21650R2

Dear Dr. Wang,

We’re pleased to inform you that your manuscript has been judged scientifically suitable for publication and will be formally accepted for publication once it meets all outstanding technical requirements.

Kind regards,

Jitendra Yadav, Ph.D.

Academic Editor

PLOS ONE

Reviewers' comments:

Reviewer's Responses to Questions

**Comments to the Author**

1. If the authors have adequately addressed your comments raised in a previous round of review and you feel that this manuscript is now acceptable for publication, you may indicate that here to bypass the “Comments to the Author” section, enter your conflict of interest statement in the “Confidential to Editor” section, and submit your "Accept" recommendation.

Reviewer #2: All comments have been addressed

Reviewer #4: All comments have been addressed

2. Is the manuscript technically sound, and do the data support the conclusions?

Reviewer #2: Yes

Reviewer #4: Yes

3. Has the statistical analysis been performed appropriately and rigorously? 

Reviewer #2: Yes

Reviewer #4: Yes

4. Have the authors made all data underlying the findings in their manuscript fully available?

Reviewer #2: Yes

Reviewer #4: Yes

5. Is the manuscript presented in an intelligible fashion and written in standard English?

Reviewer #2: Yes

Reviewer #4: Yes

6. Review Comments to the Author

Reviewer #2: The authors took seriously my previous comments and made the required revisions in the manuscript. The quality and presentation of the manuscript have been improved. Therefore, I recommend acceptance.

Reviewer #4: This paper has substantially addressed the comments made in the review and is acceptable for publication; the manuscript is technically sound and provides data to support the conclusions; and the language of the submitted manuscript is clear, correct, and unambiguous.

7. PLOS authors have the option to publish the peer review history of their article (what does this mean?). If published, this will include your full peer review and any attached files.

Reviewer #2: No

Reviewer #4: No

---

## [Editor Report · Acceptance letter]

18 Mar 2024

PONE-D-23-21650R2 

PLOS ONE

Dear Dr. Wang, 

I'm pleased to inform you that your manuscript has been deemed suitable for publication in PLOS ONE. Congratulations! Your manuscript is now being handed over to our production team.

Kind regards, 

on behalf of

Dr. Jitendra Yadav 

Academic Editor

PLOS ONE